# Boundary conditions for the Middle Miocene Climate Transition (MMCT v1.0)

Amanda Frigola[1], Matthias Prange[1,2], Michael Schulz[1,2]

[1]MARUM Center for Marine Environmental Sciences, University of Bremen, Bremen, 28359, Germany

[2]Faculty of Geosciences, University of Bremen, Bremen, 28359, Germany

*Correspondence to*: Amanda Frigola (afrigola@marum.de)

**Abstract.** The Middle Miocene Climate Transition was characterized by major Antarctic ice-sheet expansion and global cooling during the interval ~15–13 Ma. Here we present two sets of boundary conditions for global general circulation models characterizing the periods before (Middle Miocene Climatic Optimum; MMCO) and after (Middle Miocene

Glaciation; MMG) the transition. These boundary conditions include Middle Miocene global topography, bathymetry and vegetation. Additionally, Antarctic ice volume and geometry, sea-level and atmospheric $CO_2$ concentration estimates for the MMCO and the MMG are reviewed. The MMCO and MMG boundary conditions have been successfully applied to the Community Climate System Model version 3 (CCSM3) to provide evidence of their suitability for global climate modeling. The boundary-condition files are available for use as input in a wide variety of global climate models and constitute a

valuable tool for modeling studies with a focus on the Middle Miocene.

## 1 Introduction

The Middle Miocene (ca. 16–11.6 Ma) was marked by important changes in global climate. The first stage of this time period, the Middle Miocene Climatic Optimum (MMCO), was characterized by warm conditions, comparable to those of the late Oligocene. Although global climate remained warmer than present-day during the whole Miocene (Pound et al., 2012),

an important climate transition associated with major Antarctic ice-sheet expansion and global cooling took place between ~15 and 13 Ma, the so called Middle Miocene Climate Transition (MMCT). Major evidence for this transition is the increase in $\delta^{18}O$ shown in benthic foraminiferal records (e.g. Lear et al., 2010; Shevenell et al., 2008; Holbourn et al., 2005).

An increase in benthic foraminiferal $\delta^{18}O$ reflects either an increase in global ice volume, a decrease in bottom water temperature (BWT), or a combination of both. Different studies using benthic foraminiferal Mg/Ca ratios (an independent

proxy for BWT) to separate the global ice volume from BWT signals in the benthic foraminiferal $\delta^{18}O$ records conclude that both an increase in global ice volume and a decrease in BWT occurred during the MMCT, though global ice volume was the main part (65–85%) in the benthic $\delta^{18}O$ signal (Lear et al., 2010; Lear et al., 2000; Shevenell et al., 2008).

Mg/Ca studies indicate that the cooling of bottom waters across the MMCT was within a range of ~0.5 to ~3°C (Lear et al., 2010; Lear et al., 2000; Shevenell et al., 2008; Billups and Schrag, 2002; Billups and Schrag, 2003).

Studies by Kominz et al. (2008) and Haq et al. (1987), based on backstripping techniques, and John et al. (2011), combining backstripping techniques with benthic foraminiferal $\delta^{18}O$, indicate an important eustatic sea-level fall across the MMCT (see Sect. 3), providing further evidence of ice-sheet expansion. Lewis et al. (2007) present data from glacial deposits in Southern Victoria Land (East Antarctica) showing local ice-sheet expansion at different time intervals between ~13.85 and ~12.44 Ma. They state that this ice-sheet expansion was preceded by significant atmospheric cooling, with glacial deposits showing

evidence of a permanent shift from wet to cold in the thermal regime of local glaciers at ~13.94 Ma. Levy et al. (2016) analyze data from the ANDRILL–2A drill site, situated in the western Ross Sea, ~30 km off the coast of Southern Victoria Land. The ANDRILL record presents two unconformities spanning the intervals ~15.8 to ~14.6 Ma and ~14.4 Ma to Late Miocene. These unconformites are interpreted to be caused by local episodes of grounded ice advance eroding material at the site at different times within those two intervals. A global compilation of paleobotanical data by Pound et al. (2012) shows

cooling and/or drying in some mid-latitude areas across the MMCT, suggesting that this transition did not only affect high latitudes.

The causes for the MMCT are a matter of debate. Suggested driving mechanisms for this transition include a drop in atmospheric $pCO_2$, changes in ocean circulation and water masses driven by ocean gateways reconfiguration and/or orbitally triggered atmospheric heat and moisture transport variations (Flower and Kennett, 1994; Holbourn et al., 2007; Holbourn et

al., 2005). Langebroek et al. (2010), for example, using an isotope enabled ice-sheet–climate model forced with a $pCO_2$ decrease and varying time-dependent orbital parameters, modeled an increase in $\delta^{18}O$ of sea water in good agreement with published MMCT estimates.

Our aim is to assemble Middle Miocene boundary conditions for global coupled General Circulation Models (GCMs), setting up an improved basis to investigate the MMCT from a modeling perspective. The boundary conditions include global

topography, bathymetry and vegetation for the Middle Miocene (see Sect. 9). Besides, Antarctic ice volume and geometry, sea-level and atmospheric $CO_2$ concentration estimates for the periods before (Middle Miocene Climatic Optimum, MMCO) and after (Middle Miocene Glaciation, MMG) the MMCT are reviewed and their uncertainties are discussed. The global topography and bathymetry presented here are mainly based on the Middle Miocene reconstruction by Herold et al. (2008), which has already been used in previous modeling studies (e.g. Herold et al., 2012; Herold et al., 2011; Krapp and Jungclaus,

2011). However, we implement some important modifications with regard to Antarctic ice-sheet geometry, sea-level and configuration of the South East Asian and Panama seaways taking into account recent reconstructions. Vegetation cover used in most previous Middle Miocene modeling studies with prescribed vegetation was mainly based on Wolfe's (1985) Early Miocene reconstruction (e.g. Herold et al., 2011; Tong et al., 2009; You et al., 2009). Here, also Middle Miocene data (Pound et al., 2012; Morley, 2011) have been used.

Our study provides the core boundary conditions required to set up GCM experiments with a Middle Miocene configuration. With this configuration as a starting point, a wide variety of sensitivity studies with a focus on the Middle Miocene and its global climate transition can be performed. Despite the relatively low availability of Middle Miocene data, our assemblage of boundary conditions reflects the state-of-the-art in Miocene research. Results from two model runs with the Community Climate System Model version 3 (CCSM3) (Collins et al., 2006) using the MMCO and MMG boundary conditions are also
presented.

## 2 Antarctic ice-sheet geometry

Giving quantitative Antarctic ice volume estimates for the MMCT is at best challenging. Most sediment core studies present ice volume estimates in units of sea water $\delta^{18}O$. Backstripping methods provide sea-level rather than ice volume estimates. More direct Antarctic ice volume estimates can be derived from modelling studies (Gasson et al., 2016; Langebroek et al.,
2009; Oerlemans 2004).

Gasson et al. (2016) performed a series of simulations with an ice-sheet model asynchronously coupled to a regional climate model and an isotope-enabled GCM using Middle Miocene palaeogeography and a range of atmospheric $CO_2$ concentrations and extreme astronomical configurations. The study tested two different Antarctic bedrock topography scenarios: one scenario with present day bedrock topography and the other one with an approximate Middle Miocene bedrock topography.
For a range of $CO_2$ concentrations between 500 and 280 ppmv and changing orbital parameters ("warm astronomical configuration" versus "cold astronomical configuration") an increase in Antarctic ice volume from 11.5 (17.2) million km³ in the warmer climate to 26.7 (35.5) million km³ in the colder climate was simulated using modern (Middle Miocene) bedrock topography.

Langebroek et al. (2009) used a coupled ice-sheet–climate model forced by atmospheric $CO_2$ and insolation changes to
reconstruct Antarctic ice volume across the MMCT. The experiment with the best fit to $\delta^{18}O$ data was forced by a $CO_2$ drop from 640 ppmv to 590 ppmv at around ~13.9 Ma and simulates an increase in Antarctic ice volume from ~6 million to ~24 million km³ across the MMCT.

Oerlemans (2004) derived Cenozoic Antarctic ice volume variations by means of a simple quasi-analytical ice-sheet model and two different $\delta^{18}O$ benthic foraminiferal records. The ice-sheet model approximates Antarctic ice volume as a function of
deep sea temperature. Using the Zachos et al. (2001) benthic foraminiferal $\delta^{18}O$ data, an ice volume increase from ~5 million to ~23 million km³ for the MMCT was obtained, while for the $\delta^{18}O$ curve by Miller et al. (1987) the increase was only from ~15 million to ~23 million km³.

Mainly based on the studies by Oerlemans (2004) and Langebroek et al. (2009), we set a total Antarctic ice-sheet volume of 23 million km³ for the MMG (Table 1; Fig. 1). This value is within the range of published estimates, although smaller than

the values estimated by Gasson et al. (2016) in their "cold climate" experiments with extreme astronomical configuration. For the MMCO we assume a total Antarctic ice-sheet volume of 6 million km³. This volume estimate is in good agreement with the values given by Langebroek et al. (2009) and Oerlemans (2004), although significantly lower than Gasson et al. (2016) (see above).

In this study, we opted for using ice-sheet model-derived Antarctic topography estimates from an earlier Cenozoic time period with similar Antarctic ice volume. These data were kindly provided by David Pollard (Pennsylvania State University) and correspond to the Oligocene Oi-1 glaciation event. They were obtained with a model version close to that described in Pollard and DeConto (2012), but with no marine ice physics, so that any floating ice is immediately removed. The bedrock-elevation boundary conditions used in that run are from the modern ALBMAPv1 dataset (Le Brocq et al., 2010). Climate forcing is obtained from a matrix of GCM climates for various orbits, $CO_2$ levels and ice sizes (Pollard, 2010). Insolation is based on Laskar et al. (2004). The run is 12 Myr long, nominally from "37 Ma to 25 Ma". The configuration we used to represent MMCO conditions corresponds to 34.8 Ma; the one representing MMG conditions, to 33 Ma.

In the MMG configuration the whole East Antarctica is covered with a single ice-sheet meanwhile the islands of West Antarctica contain only some small ice caps, with no marine-based ice-sheets. In the MMCO configuration Antarctica is only partially glaciated, with an ice cap covering the Transantarctic Mountains and another one over eastern East Antarctica. Again, West Antarctica contains only some ice caps, with no marine-based ice-sheets (Fig. 1). We deem the ice configurations from the Oi-1 glaciation run appropriate for the MMCT as they match total ice volumes of 23 million km³ for the MMG and of 6 million km³ for the MMCO, values that are within the range of published ice volume estimates for the MMG and MMCO, respectively. Besides, there is little to link Pollard's run to a specific time period (except for the Laskar orbits). The configurations of Oerlemans (2004) and Langebroek et al. (2009) were discarded in our study because they are rather simple (no two-dimensional representation of the Antarctic ice-sheet is available from those studies). The configuration of Gasson et al. (2016) could be considered in future sensitivity studies, since uncertainties in ice volume estimates are high. Nevertheless, the distribution of ice in our study is comparable to that in Gasson et al. (2016): for the MMG, a continental-scale ice-sheet exists in East Antarctica with ice thicknesses of ~3000-4000 m, although in West Antarctica there is less ice in Pollard's data; for the MMCO, the ice-sheets occupy similar positions, although they are less extensive in Pollard's data.

The ice volume values of 6 million and 23 million km³ for the MMCO and the MMG, respectively, imply an increase of 17 million km³ across the MMCT. This value is within the range of published estimates and in reasonable agreement with the values by Langebroek et al. (2009), Oerlemans (2004) as well as Gasson et al. (2016) (see above).

Although some studies suggest ice would have been present in the Northern Hemisphere during the Middle Miocene (Thiede et al., 2011; DeConto et al., 2008), little is known about its temporal and spatial distribution. In view of the lack of concrete data, Northern Hemisphere ice was neglected in the current study.

## 3 Sea-level change across the MMCT

For the sake of coherency within the current set of Middle Miocene boundary conditions, we opted for a sea-level change
across the MMCT that is consistent with our ice-sheet volume estimates for the MMCO and MMG (see previous Sect.), presuming that global sea-level change across the MMCT time interval was dominated by glacio-eustasy. The ice-sheet volume estimates were converted into sea-level equivalents according to the following equation (Langebroek et al., 2009):

$$S_{eq} = ( \rho_{ice} * V_{ice}) / (\rho_{water} * A_0) \qquad (1)$$

where $S_{eq}$ represents sea-level equivalent, $V_{ice}$ ice-sheet volume, $\rho_{ice}$ density of ice, $\rho_{water}$ density of water and $A_0$ ocean
surface area. With $\rho_{ice} = 910$ kg/m³, $\rho_{water} = 1000$ kg/m³ and $A_0 = 360.5$ million km² (present-day approx.), values of 16 and 59 m for the MMCO and the MMG respectively were obtained (as in Langebroek et al., 2009), and thus a sea-level fall across the MMCT of 43 m (Table 1).

Lear et al. (2010) combine Mg/Ca and Li/Ca ratios to estimate BWT at ODP Site 761 across the MMCT. Sea water $\delta^{18}O$ is then derived by extracting the BWT signal from the $\delta^{18}O$ signal from the same site. The data suggest an increase in sea water
$\delta^{18}O$ of 0.6 per mil between 15.3 and 12.5 Ma. This value is converted into a sea-level fall equivalent using the Pleistocene calibration of 0.08–0.11 per mil per 10 m sea-level (Fairbanks and Matthews, 1978), obtaining an eustatic sea-level drop of ~55–75 m between 15.3 and 12.5 Ma.

De Boer et al. (2010) used one-dimensional ice-sheet models and benthic foraminiferal $\delta^{18}O$ data (Zachos et al., 2008) to derive eustatic sea-level and BWT changes over the last 35 Ma. In their approach, surface air temperature has been derived
through an inverse procedure from the benthic $\delta^{18}O$ record. The study suggests a sea-level drop of ~40 m between 15 and 12 Ma, i.e., from ~55 to ~15 m above present-day.

Kominz et al. (2008) combine data from different boreholes collected from the New Jersey and Delaware Coastal Plains to derive sea-level for the last 108 Ma through backstripping. They register a ~20 m sea-level fall between ~14.2 and ~12.8 Ma, i.e., from ~25 to ~5 m above present day. These data contain some unconformities however, implying that the actual
amplitude of the MMCT sea-level drop could have been indeed higher than proposed there.

John et al. (2011) estimate Middle Miocene sea-level changes based on backstripping and benthic foraminiferal oxygen isotope data. Backstripping is applied to sediment core data from the Marion Plateau, offshore Northeastern Australia, obtaining a range of sea-level variations which is then further constrained using benthic foraminiferal $\delta^{18}O$ data from Zachos

et al. (2001). John et al. (2011) suggest a 53–69 m sea-level fall between 16.5 and 13.9 Ma. Unfortunately, their analyses are limited to this time interval.

Our sea-level fall estimate of 43 m across the MMCT is consistent with Langebroek et al. (2009) (~43 m) and in good agreement with the study by De Boer et al. (2010) (~40 m), although somewhat larger than the values by Gasson et al. (2016) (~30–36 m) and considerably higher than Kominz et al. (2008) estimate (~20 m of sea-level fall). By contrast, John et al. (2011) suggest a higher amplitude of sea-level drop (~53–69 m), similar to the ~55–75 m by Lear et al. (2010). As such, our assumption of 43 m sea-level fall lies well within the range of published estimates.

## 4 Atmospheric CO$_2$ concentration

Numerous studies based on both marine (Foster et al., 2012; Pearson and Palmer, 2000; Tripati et al., 2009; Pagani et al., 2005) and terrestrial (Retallack, 2009; Kuerschner et al., 2008) proxies reconstructing Middle Miocene atmospheric CO$_2$ levels are present in the literature. However, the range of published estimates is rather wide. The difference in the CO$_2$ estimates between the various studies arises most likely from method-related uncertainties and/or the relatively coarse temporal resolution of some of the datasets. Based on planktonic foraminiferal boron isotopic data, Foster et al. (2012) suggest atmospheric pCO$_2$ values reaching a maximum of ~390 ppmv at ~15.8 Ma and decreasing to ~200 ppmv by ~12 Ma. Pearson and Palmer (2000), using the same method, obtain a maximum value of ~300 ppmv at ~16.2 Ma followed by a decline to ~140 ppmv by ~14.7 Ma and an increase to ~225 ppmv by ~11.4 Ma. Tripati et al. (2009), by means of surface-dwelling foraminiferal boron/calcium ratios, obtain a maximum value of ~430 ppmv at ~15.1 Ma followed by a decrease to ~340 ppmv by ~12 Ma and down to ~230 ppmv by ~9.9 Ma. Retallack (2009), based on carbon isotopes of pedogenic carbonate, suggests atmospheric pCO$_2$ levels reaching a maximum of ~850 ppmv at ~15.6 Ma, dropping down to ~115 ppmv by ~14.6 Ma and increasing to ~430 ppmv by ~12.8 Ma. Kuerschner et al. (2008) used stomatal-density data from fossil leaves to support pCO$_2$ values reaching a maximum over ~400–500 ppmv at ~15.5 Ma, decreasing to ~280 ppmv by ~14 Ma and increasing to ~340 ppmv by ~12 Ma. By contrast, Pagani et al. (2005), using a method based on alkenone carbon isotopes, obtain pCO$_2$ values oscillating between ~200 ppmv and ~300 ppmv across the Middle Miocene.

We chose atmospheric CO$_2$ concentrations of 400 ppmv and 200 ppmv to represent the MMCO and the MMG respectively (Table 1). Although somewhat arbitrary, these values are within the range of published estimates. The 400 ppmv MMCO is in favourable agreement with Foster et al. (2012) (~392 ppmv at ~15.8 Ma) and Tripati et al. (2009) (~430 ppmv at ~15.1 Ma), although higher than Pearson et al. (2000) (~300 ppmv at ~16.2 Ma) and Pagani et al. (2005) (~300 ppmv at ~15 Ma), and lower than Kürschner et al. (2008) (> ~400-500 ppmv at ~15.5 Ma) and Retallack (2009) (~852 ppmv at ~15.6 Ma) maxima. The 400 ppmv estimate is also in good agreement with the most recent alkenone- and boron isotope-based pCO$_2$ reconstructions for the MMCO by Zhang et al. (2013) and Greenop et al. (2014). The 200 ppmv MMG estimate is in good

agreement with Foster et al. (2012) (~200 ppmv at ~12 Ma) and Pagani et al. (2005) (~200 ppmv at ~13 Ma), although

higher than Pearson et al. (2000) (~140 ppmv at ~14.7 Ma) and Retallack (2009) (~116 ppmv at ~14.6 Ma), and lower than Tripati et al. (2009) (~340 ppmv at ~12 Ma) and Kürschner et al. (2008) (~280 ppmv at ~14 Ma) minima.

## 5 Global topography and bathymetry

We present here two different global Middle Miocene topography/bathymetries, characterizing the MMCO and the MMG periods (Fig. 2, Table 1, Sect. 9). Both global topography/bathymetries are mainly based on the 2°x2° reconstruction of

Herold et al. (2008), although some important modifications, which will be described below, were applied to their original dataset. To reconstruct paleogeography and paleotopography Herold et al. (2008) used a global plate rotation model and geological data. Ocean depth was reconstructed by applying an age–depth relationship to a global Middle Miocene isochron map. Sediment thickness and large igneous provinces were also considered in the reconstruction of ocean depth.

In Herold et al. (2008), the Andes and the Tibetan plateau were set with estimated Early to Middle Miocene elevations. The

Rocky Mountains and the east African topography were reduced to 75 and 25% of their current elevation, respectively. The Greenland topography was also reduced, representing ice-free conditions. The Bering Strait was closed and the Hudson Bay removed. Unlike Herold et al. (2008), who assumed the Tethys seaway closed, we decided to leave the seaway open. According to Rögl (1999), the Tethys passage closed during the Burdigalian (20.44–15.97 Ma), re-opened temporary during the Langhian (15.97–13.65 Ma) and closed again during the Serravallian (13.65–11.62 Ma). Also based on Rögl (1999), the

Paratethys was intermittently connected and disconnected from the global ocean during the Burdigalian to Serravallian interval. In Herold et al. (2008), as in our reconstruction, the Paratethys is connected to the global ocean. In view of the variable configuration of the Tethys/Paratethys across the Middle Miocene, it would be recommendable to test different Tethys/Paratethys configurations when performing Middle Miocene experiments with GCMs, although such testing can be limited by model constraints: seas disconnected from the global ocean can produce freshwater imbalance in GCMs and

narrow ocean passages require high resolution ocean grids to allow ocean flux calculation (Rosenbloom et al., 2011). Some studies, indeed, suggest that the Tethys passage closure might have played a role in Middle Miocene atmosphere and ocean circulation patterns (Ramstein et al., 1997; Hamon et al., 2013).

Additional important modifications were applied to the original dataset of Herold et al. (2008) regarding Antarctic ice-sheet geometry, sea-level and configuration of the South East Asian and Panama seaways (see discussion below in subsections

5.1–5.4) (Fig. 3).

## 5.1 Antarctica

Antarctic ice-sheet geometry was modified for consistency with our ice-sheet volume estimates for the MMCO and MMG (see Sect. 2). In the high southern latitudes, from 60°S to 90°S, the original topography/bathymetry from Herold et al. (2008) was replaced by the Antarctic topography/bathymetry data from David Pollard described above (Fig. 1e,f).

## 5.2 Sea-level

Sea-level was adjusted to ~48 m and ~5 m above present-day for the MMCO and MMG, respectively (Table 1). These values were derived from our sea-level equivalent estimates (see above) by assuming 64 m of present-day sea-level equivalent. This present-day estimate is in good agreement with Vaughan et al. (2013) (58.3 m for the Antarctic ice-sheet and 7.36 m for the Greenland ice-sheet). The adjustments were applied taking into account that sea-level is ~52 m above present-day in Herold et al. (2008) (Nicholas Herold, pers. comm.) and at present-day level in David Pollard's data. We note that the sea-level adjustment of 4 m (i.e. 48 m above present-day instead of 52 m) with respect to Herold et al. (2008) for the MMCO is minor and has virtually no effect in common global climate models at low spatial resolution.

## 5.3 South East Asian gateway

South East Asian paleogeography was modified based on Hall's (2012) reconstruction constrained at 15 Ma (Fig. 4). Hall's data, available as a georeferenced image, were converted into grid format using ArcGIS. Qualitative height/depth values were assigned to the different geographic features: ~2800 m for volcanoes, ~1000 m for highlands, ~250 m for land, ~-22 m for carbonate platforms, ~-200 m for shallow sea, <-4000 m for deep sea, and ~-5500 m for trenches. After embedding the data into the MMCO global dataset, minor manual smoothing was applied at the margins of the embedded region. Here, shallow bays were removed and single, shallow grid points surrounded by much deeper grid points were deepened to the adjacent depth. In total, these modifications affected ~0.5% of the total number of grid points.

Based on the depth values assigned to the different geographic features in the South East Asia reconstruction, the deepest connection between the Indian and the Pacific Ocean would be only ~200 m deep. This is probably too shallow, since in the Middle Miocene the Indonesian gateway was likely open for both surface and intermediate water (Kuhnt et al., 2004). Deep water passages were therefore added to the MMCO global topography/bathymetry dataset, based on the postulated Middle Miocene ocean paths across the Indonesian archipelago described in Kuhnt et al. (2004) (Fig. 4). The depth values assigned to these passages are shallower than those of the present-day passages (Kuhnt et al., 2004). Based on an educated guess the depths of the two eastern passages were set to 800 m, shallower than that of the present-day eastern Lifamatola (~1940 m) and Timor (~1300–1450 m) passages (Gordon et al., 2003). The northwestern passage was assigned a depth of 1000 m, shallower than that of the present-day northwestern Sangihe Ridge sill (~1350 m) (Gordon et al., 2003). The depth of the

southwestern passage was set to 600 m, slightly shallower than that of the present-day southwestern Dewakang sill (680 m) (Gordon et al., 2003). Assigned widths are somewhat arbitrary. The eastern passages were both given a width of ~150 km (~3 grid cells), and the northwestern and southwestern passages a width of ~350 km (~6 grid cells) and ~220 km (~4 grid cells) respectively.

For the MMG, the same South East Asia topography/bathymetry reconstruction was used as for the MMCO after applying a
~43 m sea-level change, in correspondence with our MMCT sea-level fall estimate (see above).

## 5.4 Panama seaway

Also the Panama seaway was modified such that substantial differences exist compared to Herold et al. (2008) original dataset. Montes et al. (2012) suggest a narrow Panama strait already by the Early Miocene, with a width of ~200 km. Therefore, the width of the seaway was set accordingly in our paleogeographic dataset, together with a depth of ~1000 m
which reflects Panama sill reconstructions for the Middle Miocene by Duque-Caro (1990).

## 6 Global vegetation

The current MMCO and MMG global vegetation reconstructions (Fig. 5, Sect. 9) were based on Pound et al. (2012), Wolfe (1985) and Morley (2011). The study by Pound et al. (2012) includes global vegetation reconstructions for the Langhian (15.97–13.65 Ma) (approximately the end of the MMCO) and the Serravallian (13.65–11.61 Ma) (roughly coincident with
the MMG). Wolfe's (1985) study contains an Early Miocene global vegetation reconstruction. The reconstructions from Pound et al. (2012) and Wolfe (1985) are the only Early/Middle Miocene global reconstructions. Morley's (2011) Middle Miocene reconstruction focuses only on the distribution of tropical forests.

The study by Pound et al. (2012), based on paleobotanical evidence from 617 Middle/Late Miocene data locations, constituted the main source of global vegetation data for our MMCO and MMG boundary conditions. Morley's (2011) study
added some detail to the current reconstructions in the tropical areas. Wolfe's (1985) Early Miocene data were used in regions where the Middle Miocene reconstructions by Pound et al. (2012) and Morley (2011) had scarce data coverage and also to characterize the vegetation patterns of the main mountain ranges.

The above studies use different nomenclatures to classify the different types of vegetation. Pound et al. (2012) use the BIOME4 classification (Kaplan, 2001), while Wolfe (1985) uses the classification scheme described in Wolfe (1979). Here,
we coded the data in the Land Surface Model (LSM) (Bonan, 1996) biome classification scheme. All biome names in this study are in italics, with LSM biome names in addition set in quotation marks. Table 2 shows how the data from Pound et al. (2012), Wolfe (1985) and Morley (2011) were converted into the LSM scheme. However, we note that the correspondence between biomes of different schemes is not always optimal (see notes in Table 2). The *warm-temperate evergreen broadleaf*

*and mixed forest* BIOME4 biome was converted into the LSM scheme as "*warm mixed forest*". The "*warm mixed forest*"

LSM biome represents a mixture of needleleaf evergreen temperate trees and broadleaf deciduous temperate trees, meanwhile the *warm-temperate evergreen broadleaf and mixed forest* BIOME4 biome represents either a) temperate broadleaf evergreen trees alone, or b) cool conifer trees mixed with temperate broadleaf evergreen trees, or c) temperate deciduous trees mixed with either temperate broadleaf evergreen trees or cool conifer trees. The conversion is suboptimal because no broadleaf evergreen temperate trees are present in the "*warm mixed forest*" LSM biome. Nevertheless, the

"*warm mixed forest*" still constitutes a fair representation of the *warm-temperate evergreen broadleaf and mixed forest* BIOME4 biome. In case a more precise representation in the LSM scheme of the *warm-temperate evergreen broadleaf and mixed forest* BIOME4 biome was required, the LSM scheme could be modified by adding a new biome containing the exact specific plant types present in the *warm-temperate evergreen broadleaf and mixed forest* BIOME4 biome described above (a similar approach is used in Herold et al., 2010). Note also that Table 2 does not contain all the biomes present in the different

vegetation schemes, but only the ones effectively used in the current reconstructions. All biome names in the following are italized, with LSM biome names in addition set in quotation marks.

Neither Pound et al. (2012) nor Wolfe (1985) or Morley (2011) provide the data in a gridded format, a requirement for the use in GCMs. According to the geographical coordinates specified in those reconstructions, we merged the data on a 2°x2° latitude/longitude grid for MMCO and MMG, giving preference to the Middle Miocene reconstructions by Pound et al.

(2012) and Morley (2011). The Early Miocene reconstruction of Wolfe (1985) was only used at locations where no data are available from the other two reconstructions. An exception are the Alps, Rocky Mountains, Himalayas and Tibetan Plateau, where we used the information from Wolfe (1985). At the regions where the reconstructions from Pound et al. (2012) and Morley (2011) were conflicting with each other, the reconstruction of Pound et al. (2012) was used, except for locations where the latter was less well constrained by proxy information than Morley (2011).

The MMCO and the MMG global vegetation reconstructions presented here solely differ in terms of *"tundra"* distribution on Antarctica (Fig. 5). In both reconstructions *"tundra"* was assigned to the ice-free regions of Antarctica, taking account of the MMCO and MMG ice-sheet geometries (Fig. 1c,d). Assuming the *"tundra"* distribution to be the only difference in terms of vegetation between the MMCO and the MMG is a simplification. Part of the *warm-temperate evergreen broadleaf and mixed forest* (*"warm mixed forest"* in the LSM scheme) present in the middle latitudes during the Langhian was replaced by

"cooler and/or drier temperate biomes" during the Serravallian (Pound et al., 2012). This cooling and/or drying trend was observed for example in areas like western North America or Europe (Pound et al., 2012). Additionally, by the Serravallian, in the tropics, some drier biomes than those present there during the Langhian had started to spread (e.g. in South East Asia) (Pound et al., 2012). Nevertheless, the Langhian and Serravallian global vegetation patterns were 'similar' in comparison with the 'markedly different biome pattern of the Tortonian from that of the Serravallian'(see Fig. 5 and Fig. 6 in Pound et

al., 2012), which justifies our simplified approach. However, for studies with a specific focus on vegetation-triggered

climatic changes across the MMCT, the user could modify our MMG vegetation dataset (LSM scheme) as follows (based on Pound et al., 2012): a) in the mid-latitudes of western North America, the *"warm mixed forest"* between 40-50°N could be partly replaced with *"warm broadleaf deciduous forest"*. Also a *"cool mixed forest"* could be added in the same region at 42°N; b) in Europe, some *"deciduous shrub land"* could be added to the *"warm mixed forest"* in southern France between 42.5-44°N and 6-9°E, and also between 38-47°N and 29-36°E.

We are aware that the Middle Miocene global vegetation reconstructions presented here are rather coarse. Still, we consider it a fair characterization of the MMCO and MMG global vegetation distributions. Grosso modo, the present reconstructions are characterized by *"cool mixed forest"* at high northern latitudes, predominantly *"warm mixed forest"* at middle latitudes, *"tropical broadleaf evergreen forest"* in the tropics, and *"tundra"* in the ice-free regions of Antarctica (Fig. 5). Compared to PI, the vegetation of the Middle Miocene represents a warmer and wetter climate. In the northern hemisphere high latitudes forests are warmer, with no forest tundra or tundra present. The mid-latitudes present warmer and wetter biomes, with e.g. less shrubland-type biomes. The tropics are wetter, with less savanna and less grasses. There is no evidence for neither a desert in northern Africa (Sahara) nor in central Asia. In the southern hemisphere high latitudes tundra is present at the MMCO and disappears after the Antarctic ice-sheet expansion at the MMG (Pound et al., 2012; Bonan et al., 2002). For a higher degree of detail, in GCMs including a dynamic vegetation component, our Middle Miocene vegetation datasets could be used to initialize the vegetation model. Another valid approach would be using the output from an offline vegetation model as boundary condition (e.g., the ones described in Henrot et al., 2017), although here our aim was to provide vegetation boundary conditions based on palaeobotanical data. A detailed discussion of the vegetation patterns we assigned to each region can be found in the Appendix (Sect. 10).

**7 Testing the boundary conditions with CCSM3**

To provide evidence of the suitability of our input data for global climate modeling, we applied the boundary conditions to the Community Climate System Model version 3 (CCSM3). CCSM3 is a fully coupled GCM consisting of components representing atmosphere, ocean, land and sea ice (Collins et al., 2006). A total of three runs were performed: two Miocene runs (MMCO and MMG) and a pre-industrial control run (PI). The atmosphere horizontal grid employed in the PI run, T42, is a Gaussian grid with 64 points in latitude and 128 points in longitude (~2.8° resolution). The notation T42 refers to the spectral truncation level. The land and atmosphere models share the same horizontal grid. The ocean horizontal grid, x1, is a dipole grid with 384 points in latitude and 320 points in longitude. The zonal resolution of the ocean horizontal grid is ~1°, the mean meridional resolution is ~0.5°, refined around the equator (~0.3°). The notation x1 refers to the nominal zonal resolution. The ocean and sea–ice components share the same horizontal grid. The atmosphere and ocean vertical grids have 26 and 40 vertical levels, respectively. This model grid configuration is known as T42x1. For the Miocene runs the same

grids as for the PI run were used, except for the horizontal ocean (and sea–ice) grid, for which a customized grid was used. This grid is also a dipole grid with 384 points in latitude and 320 points in longitude, although extended to ~87°S (instead of ~79°S) in order to accommodate the changes in the bathymetry off West Antarctica. In the Miocene topography/bathymetry datasets West Antarctica is for the most below sea-level, with ocean reaching down to ~85°S. If the standard CCSM3 grid had been used for the Miocene experiments, the ocean region between ~79°S and 85°S would not have been taken into account in the ocean circulation simulations. The Miocene grid was created from scratch using the CCSM3 setup tools described in Rosenbloom et al. (2011) and it is defined by the following parameters: dyeq=0.25 (meridional grid spacing at the equator, in degrees), dsig=20 (Gaussian e-folding scale at equator), and jcon=45 (rows of constant meridional grid spacing at poles). In some areas the Miocene grid presents a slightly coarser resolution than the PI grid, since both grids have the same number of grid points and the Miocene grid reaches further south than the PI grid. In order to be able to compare the Miocene and PI results, the PI model output data were regridded onto the Miocene grid using the patch recovery method (http://www.ncl.ucar.edu/Applications/ESMF.shtml), which gives better approximations than the bilinear method. We do not think interpolation has any significant effect on the results.

For the PI run, well-mixed greenhouse gases, ozone distribution, aerosols, solar constant and orbital configuration were set to PI following Otto-Bliesner et al. (2006). The boundary conditions summarized in Table 1 were used for the Miocene experiments. Well-mixed greenhouse gases, ozone, aerosols, solar constant and orbital configuration were kept the same as in PI, except for $CO_2$ (Table 3). The PI experiment was branched from the NCAR CCSM3 1870 CE control run and integrated another 150 years (850 years in total). The Miocene experiments were integrated for a total of 1500 years. The last 100 years of each simulation were used for the analysis. The temperature trends in the deep ocean (at 4-5 km depth) are < 0.14, 0.15, and 0.17 °C/100 years in the PI, MMCO, and MMG cases, respectively. At that same depth range, the salinity trends are <0.01, 0.007, and 0.01 psu/100 years for PI, MMCO, and MMG, respectively. These values represent quasi-equilibrium conditions and we consider them sufficiently small for the focus of this study.

The global mean precipitation rates are 3.00, 2.86, and 2.72 mm/day for the MMCO, MMG, and PI experiments, respectively. Some patterns distinguishing the Miocene runs from the PI run include lower precipitation rates during the Miocene along the northwest coast of South America (15°S-10°N)(up to 3-4 mm/day lower)(Fig. 6, Fig. 7), although reaching further inland. In central Africa, between 20°S-5°N, the results show values also up to 3-4 mm/day lower than PI, which might be related to the reduced Miocene east African topography (Jung et al., 2016). An intensification of the precipitation occurs in the Indian Ocean, compared to PI, with the ITCZ reaching further north at ~90°E (Bay of Bengal). On the contrary, in the equatorial Pacific the PI run exhibits higher precipitation rates than the Miocene runs. These changes in the rainbelt patterns could be related to the different extent and distribution of ice-sheets in both hemispheres. Other studies have already suggested that global ice-sheets might have an effect on the ITCZ (Holbourn et al., 2010; Groeneveld et al., 2017). Other interesting patterns are the decrease in precipitation across the MMCT over the east African coast, between 0-

20°N (0.5-2 mm/day lower), as well as the drying at high latitudes (0-1 mm/day). Regarding SSTs (Fig. 6), the highest values correspond to the MMCO experiment, with a mean global SST of 19.62°C, followed by the MMG experiment (18.04°C) and the PI experiment (16.85°C). The decrease in SST across the MMCT is particularly evident in the southern high latitudes and the northern North Pacific (values up to 6-7°C lower). When considering surface air temperatures (at 2 m height), our results show mean global values of 16.38°C, 13.88°C, and 12.16°C for the MMCO, MMG, and PI, respectively (Fig. 8). The modelled decrease in mean global surface air temperatures (2.5°C) and SSTs (1.6°C) between MMCO ($CO_2$ = 400 ppmv) and MMG ($CO_2$ = 200 ppmv) is in good agreement with the CCSM3 climate sensitivity values suggested in Kiehl et al. (2006). Our global mean surface air temperature and precipitation values support the idea of a Middle Miocene climate warmer and wetter than PI, and a cooling and drying trend across the MMCT. Mg/Ca data from ODP Hole 1171C on the South Tasman Rise indicate cooling of SSTs of ~2°C across the MMCT (Shevenell et. al, 2004). This value is within our range of cooling estimates for the Southern Ocean. Knorr and Lohmann (2014) MMCT model results show a decrease of 3.1°C in global mean surface air temperature across the MMCT, a value slightly higher than our 2.5°C estimate. Although $CO_2$ is lower in the MMG simulation than in the control run, SSTs are higher for MMG than for PI. Potential causes for a MMG climate warmer than PI are the lower extent of ice-sheets (the Antarctic ice-sheet is smaller and the northern Hemisphere free of ice-sheets in the MMG run), and the different vegetation cover (Knorr et al., 2011). However, unambiguously disentangling the effects of each of the different boundary conditions would require performing a series of sensitivity experiments, which is beyond the scope of the current study. Here our aim was testing the idoneity of the current boundary conditions as input data in GCMs for MMCO and MMG experiments.

## 8 Concluding remarks

The current study describes and provides a complete set of boundary conditions for GCMs characterizing the MMCO and the MMG periods. These boundary conditions include global topography, bathymetry and vegetation, and have a particular focus on the Antarctic ice-sheet and the South East Asian gateway. Besides, atmospheric $CO_2$ concentrations and sea-level estimates were reviewed in detail. Other GCM input data, which strongly depend on the technical details of individual GCMs, like river routing, were not discussed here, but can be constructed from the provided topographic datasets.

All data gathered in this study are available in the supplement, ready for use in GCMs. The compilation of boundary conditions for GCMs is time-consuming, especially when referring to deep time periods such as the Miocene. This assemblage of data can be used by the paleomodeling community as a base for a wide variety of Middle Miocene studies, particularly for those related to the MMCT. Output data from two global climate model experiments using these boundary conditions were briefly described here, in order to proof the suitability of the new sets of boundary conditions for modeling purposes.

Despite ongoing efforts, the Middle Miocene is still a period with scarce data availability and thus the Middle Miocene picture presented here is inevitably subject to large uncertainties. More data, improved methods and higher resolutions are required in order to improve the current boundary conditions and hence obtain more detailed and reliable model results.

## 9 Data availability

All the MMCO and MMG boundary conditions described above, as well as the CCSM3 model output files from the MMCO, MMG, and PI experiments, can be found in NetCDF format in the supplement and at PANGAEA. The MMCO and MMG global topography/bathymetry data are available from the 0.5°x0.5° lat/lon grid files mmco_topo_bathy_v1_0.nc and mmg_topo_bathy_v1_0.nc, respectively. The MMCO and MMG global vegetation data can be found in the 2°x2° lat/lon grid files mmco_veg_v1_0.nc and mmg_veg_v1_0.nc, respectively. The CCSM3 output files are named MMCO_exp.nc, MMG_exp.nc, and PI_exp.nc, for MMCO, MMG, and PI, respectively.

## 10 Appendix

### 10.1 Europe

The most widespread biome in the middle latitudes of Europe during the Langhian was the *warm-temperate evergreen broadleaf and mixed forest* (*"warm mixed forest"*) (Pound et al., 2012). On the southern and eastern coast of Spain however, drier biomes such as *temperate xerophytic shrubland* ( *"evergreen shrub land"/"deciduous shrub land"*) or *temperate deciduous broadleaf savanna* ( *"deciduous shrub land"*) were present (Pound et al., 2012) (Fig. 5). The vegetation patterns of the Serravallian were similar to those of the Langhian. However, some drying and/or cooling occurred. In east Europe, for example, the data by Pound et al. (2012) indicate the emergence of *temperate deciduous broadleaf savanna* (*"deciduous shrub land"*). These drier and/or cooler biomes developed during the Serravallian were neglected for simplification in the current reconstructions.

Evidence for the Middle Miocene European high northern latitudes (above 60°N) is missing in Pound et al. (2012). This region was painted with *mixed coniferous forest* (*"cool mixed forest"*) (Fig. 5) in the current MMCO and MMG reconstructions based on Wolfe's (1985) Early Miocene data. We also filled the Alpine area with *mixed coniferous forest* (*"cool mixed forest"*) according to Wolfe (1985).

### 10.2 Asia

The Asian high northern latitudes were assigned *"cool mixed forest"* in the current MMCO and MMG reconstructions (Fig. 5). Pound et al. (2012) suggest *cool-temperate mixed forest* (*"cool mixed forest"*) to have been the dominant biome in the

eastern Asian high northern latitudes during the Middle Miocene. No data are provided for the western Asian high northern latitudes in Pound et al. (2012). Wolfe's (1985) reconstruction suggests two different vegetation patterns for the western Asian high northern latitudes during the Early Miocene, corresponding to north and south. For simplification, only the northern pattern, consisting of *mixed coniferous forest* (*"cool mixed forest"*), was considered in the current reconstructions.

The western Asian middle latitudes were filled with *"warm mixed forest"* (Fig. 5). Pound et al. (2012) propose a "south to

north drying and cooling trend" for that region during the Middle Miocene, starting with *warm-temperate evergreen broadleaf and mixed forest* (*"warm mixed forest"*) in the southern part and ending with *temperate deciduous broadleaf savanna* (*"deciduous shrub land"*) in the northern part. For simplification, only the *warm-temperate evergreen broadleaf and mixed forest* (*"warm mixed forest"*) was considered in our current MMCO and MMG boundary conditions.

The Himalayas and the Tibetan Plateau were assigned *mixed coniferous forest* (*"cool mixed forest"*) based on Wolfe's

(1985) reconstruction.

The eastern Asian middle latitudes were mainly populated by *warm-temperate evergreen broadleaf and mixed forest* (*"warm mixed forest"*) during the Middle Miocene (Pound et al., 2012) (Fig. 5). Along the coast at ~32°N, there was a *tropical evergreen broadleaf forest* (*"tropical broadleaf evergreen forest"*), and west of 111°E a drier region containing biomes such as *temperate xerophytic shrubland* (*"evergreen shrub land"/"deciduous shrub land"*) (Pound et al., 2012). This drier region

was dismissed for simplification in our MMCO and MMG reconstructions.

*Tropical evergreen broadleaf forest* (*"tropical broadleaf evergreen forest"*) was the dominant biome in India and South East Asia during the Middle Miocene (Pound et al., 2012) (Fig. 5).

**10.3 Australia and New Zealand**

*Tropical evergreen broadleaf forest* (*"tropical broadleaf evergreen forest"*) was present in northeast Australia during the

Middle Miocene (Pound et al., 2012) (Fig. 5). Also in the east, but south of 28°S, *warm-temperate evergreen broadleaf and mixed forest* (*"warm mixed forest"*) was present (Pound et al., 2012).

In Pound et al. (2012) the description of Australia is limited to the east side. A line of *megathermal rain forest* (*"tropical broadleaf evergreen forest"*) was assigned along the north coast in the current reconstructions based on Morley's (2011) Middle Miocene data.

Wolfe's (1985) Early Miocene reconstruction suggests the vegetation patterns of east Australia to be also representative for central Australia. Assuming this would still be valid during the Middle Miocene, the *"warm mixed forest"* assigned to east Australia in the current reconstructions, was extended to central Australia.

Wolfe's (1985) data further suggest similar vegetation patterns for west Australia and the southern coast of Spain during the Early Miocene. Assuming this analogy to be still valid during the Middle Miocene, west Australia was filled with

"*evergreen shrub land*"/"*deciduous shrub land*", which is the vegetation assigned to the southern coast of Spain in the current MMCO and MMG reconstructions.

*Warm-temperate evergreen broadleaf and mixed forest* (*"warm mixed forest"*) occupied New Zealand during the Middle Miocene (Pound et al., 2012) (Fig. 5).

## 10.4 Antarctica

There is evidence for *low- and high-shrub tundra* and *prostrate dwarf-shrub tundra* (*"tundra"*) at the Antarctic margins during the Langhian (Pound et al., 2012). By the Serravallian, practically no vegetation was present on Antarctica (Pound et al., 2012). In the current MMCO and MMG reconstructions *"tundra"* was assigned to the ice-free regions, in consistence with the ice-sheet geometries described above (Fig. 5).

## 10.5 Africa and the Arabian Peninsula

Africa and the Arabian Peninsula have poor data coverage in Pound et al. (2012). Evidence from Pound et al. (2012) for the most northern part of Africa for the Middle Miocene is restricted to one site (in Tunisia), suggesting a *warm-temperate evergreen broadleaf and mixed forest* (*"warm mixed forest"*). That data site was dismissed in our current MMCO and MMG reconstructions, in view of the inappropriateness to extrapolate data from only one site to the whole surrounding region. Instead, the most northern part of Africa was set to *"evergreen shrub land"/"deciduous shrub land"* (Fig. 5). Wolfe's (1985)
Early Miocene reconstruction suggests similar vegetation patterns for that region as for the southern coast of Spain. Assuming these two regions kept similar vegetation patterns also during the Middle Miocene, *"evergreen shrub land"/"deciduous shrub land"*, the vegetation assigned to the southern coast of Spain in the current MMCO and Middle MMG reconstructions, was also assigned to the most northern part of Africa. Also a narrow belt of *megathermal rain forest* (*"tropical broadleaf evergreen forest"*) was set along the northwest coast, in agreement with Morley's (2011) Middle
Miocene reconstruction.

Madagascar was assigned *megathermal rain forest* (*"tropical broadleaf evergreen forest"*), following Morley (2011). Also based on Morley's (2011) reconstruction, an area of *megathermal rain forest* (*"tropical broadleaf evergreen forest"*) was defined along the south and southeast coast of southern Africa. Pound et al. (2012) suggest that drier tropical biomes, e.g. *tropical savanna* (*"savanna"*), were present close to the southern coast of southern Africa during the Middle Miocene. These
drier biomes were dismissed for simplification in our MMCO and MMG boundary conditions.

Still in southern Africa, north of the *"tropical broadleaf evergreen forest"* line, a region of *"warm mixed forest"* was assigned. Wolfe (1985) suggests similar vegetation patterns for that region and for southeast Australia during the Early Miocene. Assuming those two regions kept similar vegetation patterns also during the Middle Miocene, *"warm mixed*

*forest"*, the biome set for southeast Australia in the current MMCO and MMG reconstructions, was also assigned to that region.

The remaining areas of Africa and the Arabian Peninsula were assigned *"tropical broadleaf evergreen forest"*, although we are aware that this is probably too broad for a characterization, as also drier tropical biomes were present. Wolfe (1985) suggests that *tropical rain forest* (*"tropical broadleaf evergreen forest"*) and other drier tropical biomes populated that region during the Early Miocene. Pound et al. (2012) show occurrence of *tropical evergreen broadleaf forest* (*"tropical broadleaf evergreen forest"*) in equatorial Africa during the Middle Miocene, although combined with drier tropical biomes like *tropical deciduous broadleaf forest and woodland* (*"tropical broadleaf deciduous forest"*) or *tropical savanna* (*"savanna"*). On the Arabian Peninsula, a single site indicates *tropical deciduous broadleaf forest and woodland* (*"tropical broadleaf deciduous forest"*) existed in that area during the Langhian (Pound et al., 2012). These drier tropical biomes were dismissed for simplification in our boundary conditions.

## 10.6 North America and Greenland

During the Middle Miocene, the western North American high latitudes were populated with *cool-temperate mixed forest* (*"cool mixed forest"*) (Pound et al., 2012) (Fig. 5). Evidence for the eastern North American high latitudes is missing in Pound et al. (2012). Wolfe (1985) suggests two different patterns for the eastern North American high latitudes during the Early Miocene, at ~60°–65°N and north of ~65°N, respectively. For simplification only the most northern pattern, *mixed coniferous forest* (*"cool mixed forest"*), was considered in the current MMCO and MMG reconstructions.

During the Langhian, *warm-temperate evergreen broadleaf and mixed forest* (*"warm mixed forest"*) was prevalent in the western North American middle latitudes above 40°N (Pound et al., 2012). South of 40°N a drier region existed, with biomes such as *temperate xerophytic shrubland* (*"evergreen shrub land"/"deciduous shrub land"*) (Pound et al., 2012) (Fig. 5). During the Serravallian, the western North American middle latitudes became more heterogeneous in terms of vegetation, an amalgam of *warm-temperate evergreen broadleaf and mixed forest* (*"warm mixed forest"*) combined with other drier and/or cooler biomes (Pound et al., 2012). For simplification, the Langhian pattern was used in both MMCO and MMG reconstructions.

The Rocky Mountains were set to *mixed coniferous forest* (*"cool mixed forest"*) following Wolfe (1985).

The central North American middle latitudes were assigned *"warm mixed forest"* in the current MMCO and MMG reconstructions based on Wolfe (1985). That region has scarce data coverage in Pound et al. (2012) (one Langhian site, two Serravallian sites). Wolfe (1985) suggests two different patterns for the central North American middle latitudes during the Early Miocene, corresponding to south and north. For the southern part, Wolfe (1985) suggests similar vegetation patterns as for southeast Australia (assigned *"warm mixed forest"* here), and for the northern part, similar vegetation patterns as for the European middle latitudes (assigned *"warm mixed forest"* here).

However, the central North American middle latitudes were not exclusively vegetated by *"warm mixed forest"* during the Middle Miocene. Wolfe (1985) suggests the presence of "at least some interfluve grassland" in areas such as Nebraska during the late Middle Miocene. Besides, Pound et al. (2012) show some evidence for the presence of *temperate grassland* (*"cool grassland"*) (Langhian) and *temperate deciduous broadleaf savanna* (*"deciduous shrub land"*) (Serravallian) in the central American middle latitudes during the Middle Miocene. These biomes were, however, dismissed for simplification in our current reconstructions.

The eastern North American middle latitudes were also assigned *"warm mixed forest"* in the current MMCO and MMG reconstructions. Pound et al. (2012) suggest *warm-temperate evergreen broadleaf and mixed forest* (*"warm mixed forest"*) existed in eastern North America between 29°N and 39°N, both during the Langhian and the Serravallian. However, outside that interval of latitude no data were available from Pound et al. (2012). Since Wolfe (1985) suggests similar vegetation patterns for the central and eastern North American middle latitudes during the Early Miocene, assuming this analogy would still be valid during the Middle Miocene, *"warm mixed forest"* was also applied to the east, north of 39°N and south of 29°N.

No Middle Miocene data were available for Greenland from Pound et al. (2012). North Greenland was filled with *mixed coniferous forest* (*"cool mixed forest"*) in our current reconstructions based on Wolfe (1985). South Greenland was assigned *"warm mixed forest"*, given its similar latitudinal position and relative geographic proximity with Iceland, where *warm-temperate evergreen broadleaf and mixed forest* (*"warm mixed forest"*) existed during the Middle Miocene according to Pound et al. (2012).

## 10.7 Central America and south Mexico

Central America and southern Mexico were assigned *"tropical broadleaf evergreen forest"* (Fig. 5). Morley (2011) suggests *megathermal rain forest* (*"tropical broadleaf evergreen forest"*) populated that region in the Middle Miocene. Pound et al. (2012) suggest that the *tropical evergreen broadleaf forest* (*"tropical broadleaf evergreen forest"*) coexisted with drier tropical biomes (Langhian) and with temperate biomes (Langhian and Serravallian), and proposes altitude as an explanation for the presence of temperate biomes in that region during the Middle Miocene. The drier tropical biomes and the temperate biomes were dismissed in our current MMCO and MMG reconstructions for simplification.

## 10.8 Northern South America

In northern South America, the northern half and the east were filled with *"tropical broadleaf evergreen forest"* in our reconstructions (Fig. 5). Pound et al. (2012) suggest *tropical evergreen broadleaf forest* (*"tropical broadleaf evergreen forest"*) as the main biome in that region during the Langhian and Serravallian. Nevertheless, they also show evidence for some *tropical deciduous broadleaf forest and woodland* (*"tropical broadleaf deciduous forest"*) in that region during the

Serravallian. Morley (2011) suggests the presence of *megathermal rain forest ("tropical broadleaf evergreen forest")* and *monsoonal megathermal forest ("tropical broadleaf deciduous forest")* in that region during the Middle Miocene. The "*tropical broadleaf deciduous forest*" was neglected for simplification in our boundary conditions.

No data were available from Pound et al. (2012) or Morley (2011) for the southwestern part of northern South America. Within that area, the Andes were assigned *"warm mixed forest"* and the rest *"tropical broadleaf evergreen forest"*. For the Early Miocene Andes, Wolfe (1985) suggests similar vegetation patterns as for the most southern part of southern South America. Since significant uplift of the Andes would have started only in the late Miocene (Ghosh et al., 2006), we considered reasonable to assume that these regions kept similar vegetation patterns also during the Middle Miocene. In this way, *"warm mixed forest"*, the biome set in our data for the most southern part of South America (see below), was also assigned to the Andes. Surrounding the Andes there is another area with non-tropical biomes in Wolfe's (1985) reconstruction, which was dismissed here for simplification. The rest of southwest northern South America is occupied by *tropical rain forest* and *paratropical rain forest ("tropical broadleaf evergreen forest")* in Wolfe's (1985) reconstruction.

### 10.9 Southern South America

During the Middle Miocene, in the northwest of southern South America, there was a region covered by arid biomes such as *temperate xerophytic shrubland ("evergreen shrub land"/"deciduous shrub land")* (Pound et al., 2012) (Fig. 5).

In the northeast, along the coast, a narrow belt of *megathermal rain forest ("tropical broadleaf evergreen forest")* was present according to Morley's (2011) Middle Miocene reconstruction.

No evidence from Pound et al. (2012) or Morley (2011) was available for the area between the arid region in the west and the *"tropical broadleaf evergreen forest"* in the east. East of the arid region, for the area corresponding to the Andes, Wolfe (1985) proposes similar vegetation patterns for the Early Miocene as for the most southern part of southern South America. Assuming these two areas kept similar vegetation patterns also during the Middle Miocene, *"warm mixed forest"*, the biome set in the current MMCO and MMG reconstructions for the most southern part of southern South America (see below), was also assigned to that part of the Andes. For the region east of the Andes, for the Early Miocene, Wolfe (1985) proposes a vegetation pattern similar to that of southeast Australia. Assuming this analogy kept being valid also during the Middle Miocene, *"warm mixed forest"*, the biome set in the current Middle Miocene reconstructions for southeast Australia, was assigned to that region.

The south of southern South America was filled with *"warm mixed forest"* in our MMCO and MMG reconstructions (Fig. 5). Pound et al. (2012) shows evidence for *warm-temperate evergreen broadleaf and mixed forest ("warm mixed forest")* mixed with *temperate grassland ("cool grassland")* south of 35°S, and again for *warm-temperate evergreen broadleaf and mixed forest ("warm mixed forest")* at 55°S. The *temperate grassland ("cool grassland")* south of 35°S was dismissed here for simplification.

**Acknowledgements**

This research is part of the Marie Curie Initial Training Network Throughflow, funded by the E.U. 7[th] Framework Programme on Research, Technological Development and Demonstration. The CCSM3 simulations were performed on the Cray XC30/40 supercomputer of the Norddeutscher Verbund für Hoch- und Höchstleistungsrechnen (HLRN). We are particularly grateful to Robert Hall, Nicholas Herold, David Pollard and Matthew Pound for their contributions in the assemblage of boundary conditions. We also want to acknowledge Lydie Dupont, Sandra Passchier and Matthew Huber for that. We are grateful to Gary Strand, Sam Levis, Esther Brady, Stephen Yeager, and in particular to Nan Rosenbloom (NCAR) for their help in the model setup procedure. A very special thanks goes to Gabriel Gaus and Lars Nerger (HLRN) for their support in the execution of the model runs. Thanks also to Andreas Manschke for the IT support, to Ute Merkel and Gerlinde Jung for sharing their expertise in CCSM3, to Hanno Keil for his introduction to ArcGIS, and to Alexandra-Jane Henrot and Petra Langebroek for their constructive reviews that helped to improve the manuscript. We are also grateful to the Throughflow Network, and the Bremen International Graduate School for Marine Sciences GLOMAR.

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

MMCO                                                    MMG

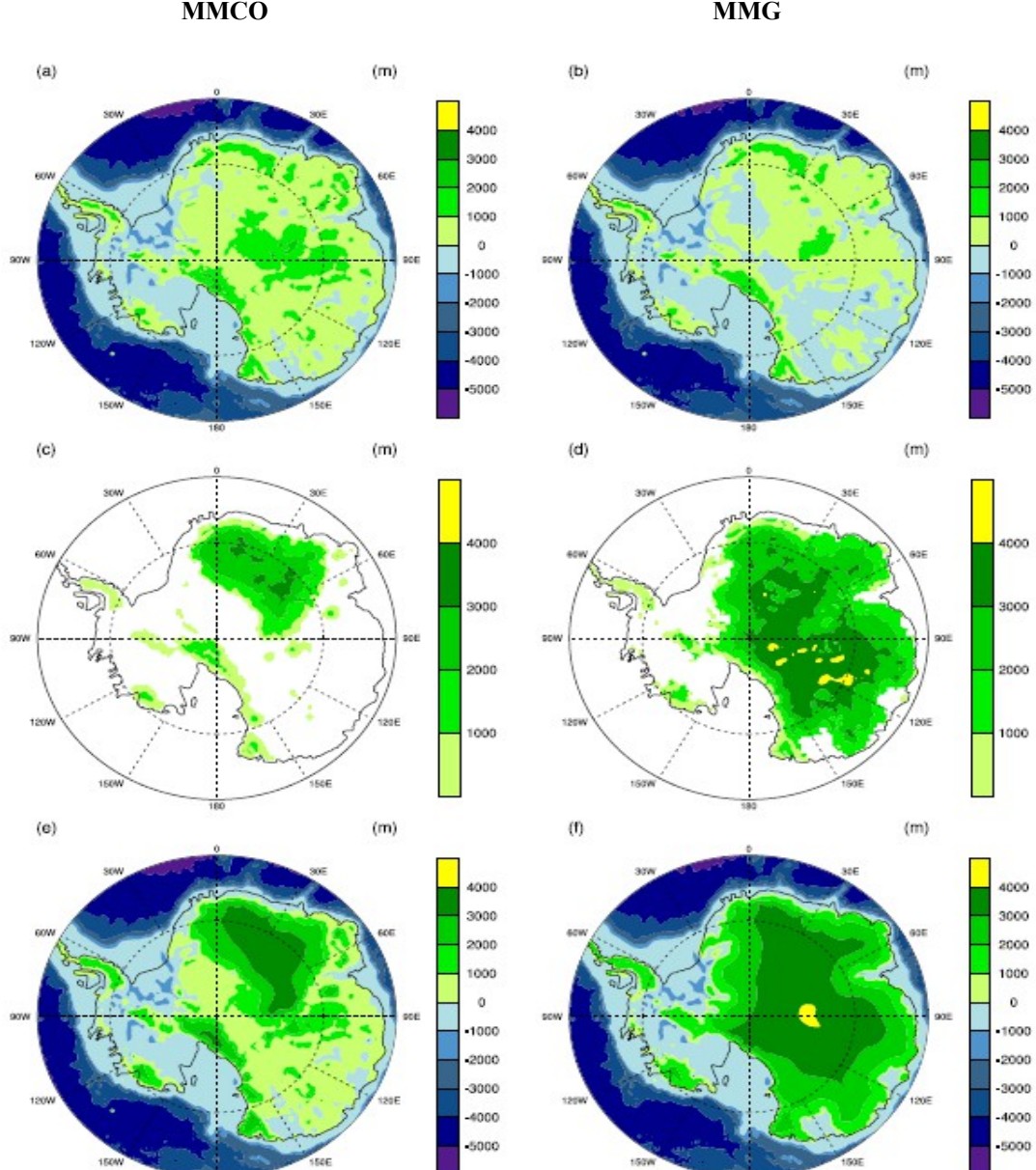

**Figure 1: Reconstruction of Antarctica for the Middle Miocene Climatic Optimum (MMCO) and the Middle Miocene Glaciation (MMG): (a), (b) bedrock elevation; (c), (d) ice thickness; (e), (f) surface elevation (bedrock elevation + ice thickness), in meters. Black lines represent present day coastline. Data from David Pollard.**


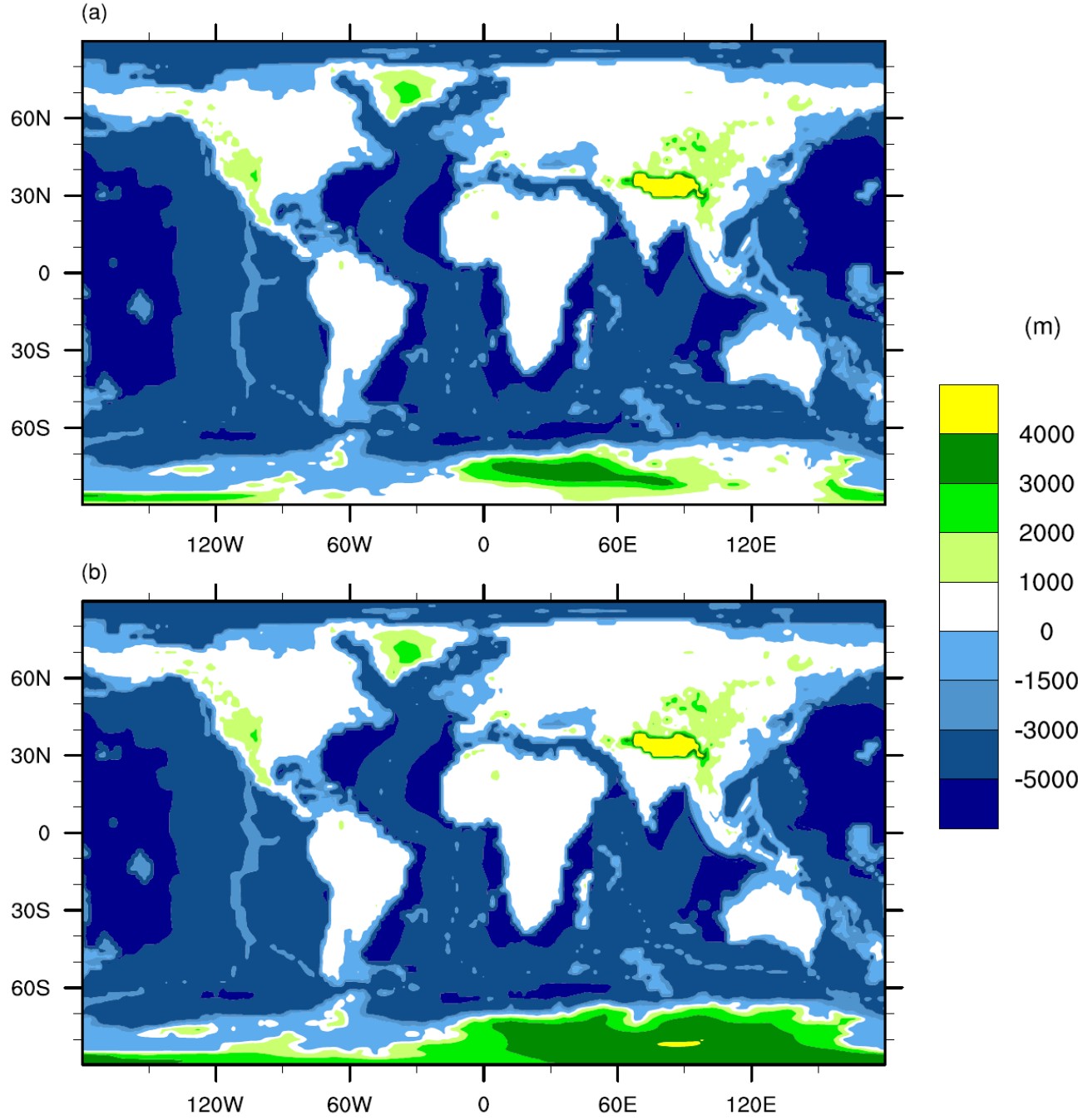

**Figure 2: Topography/bathymetry reconstruction for the (a) MMCO and the (b) MMG.**

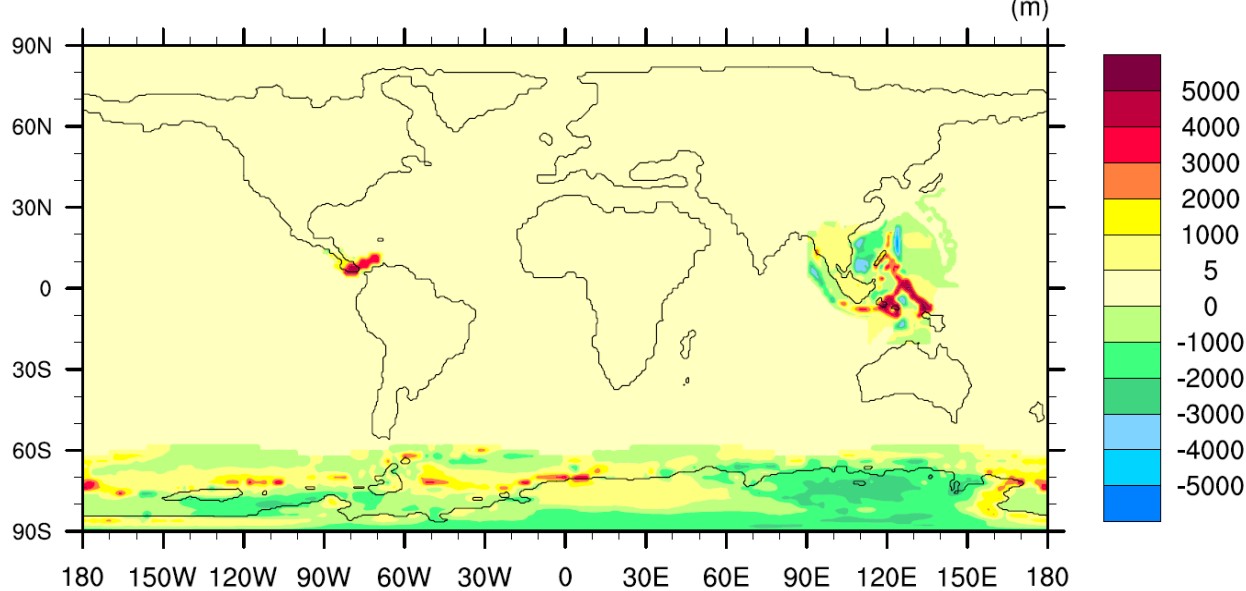

**Figure 3: Difference between MMCO and the topography/bathymetry by Herold et al. (2008), in meters. Sea level is 4 m higher in the MMCO dataset (subsection 5.2), the Indonesian Throughflow barriers are shallower (subsection 5.3), the Panama seaway is narrower (subsection 5.4), and the Antarctic topography/bathymetry is based on David Pollard's data (subsection 5.1) and consistent with MMCO ice volume estimates.**


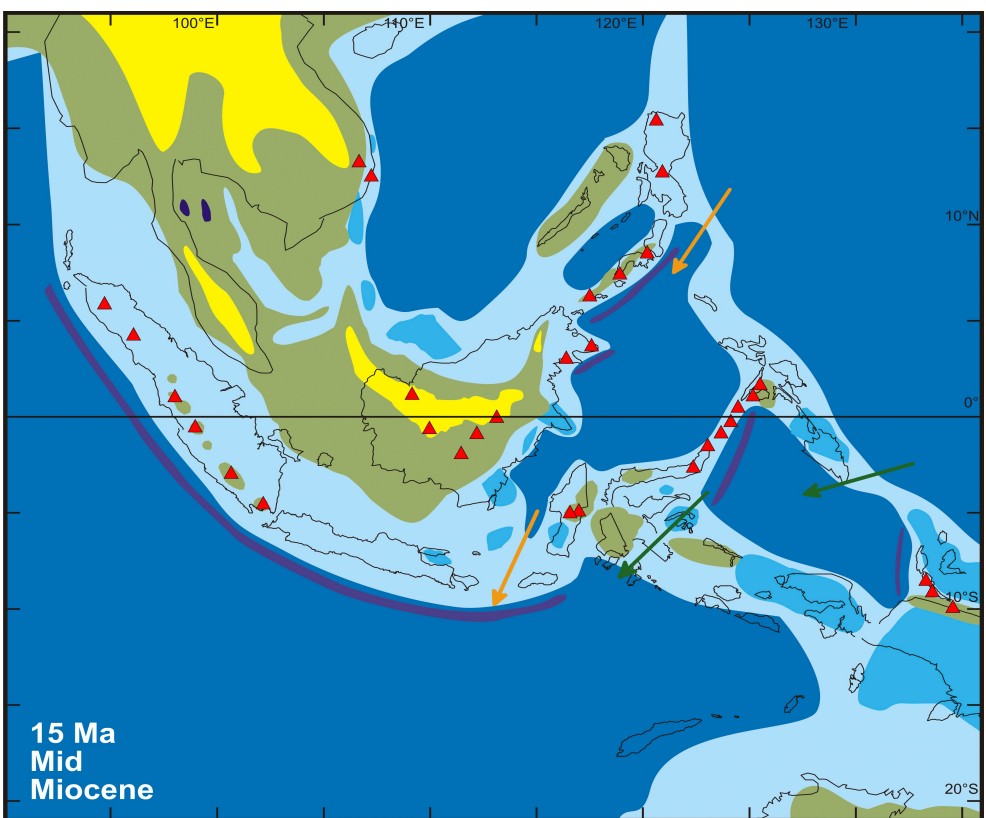

**Figure 4: Paleogeographic reconstruction of South East Asia for 15 Ma from Hall (2012). Geographic features: volcanoes in red (triangles), highlands in yellow, land in green, carbonate platforms in blue, shallow sea in light blue, deep sea in dark blue, trenches in violet. Figure courtesy of Robert Hall (Royal Holloway, University of London). Arrows represent Middle Miocene postulated ocean paths across the Indonesian archipelago and are based on Kuhnt et al. (2004). Eastern paths shown in green, western paths in orange.**



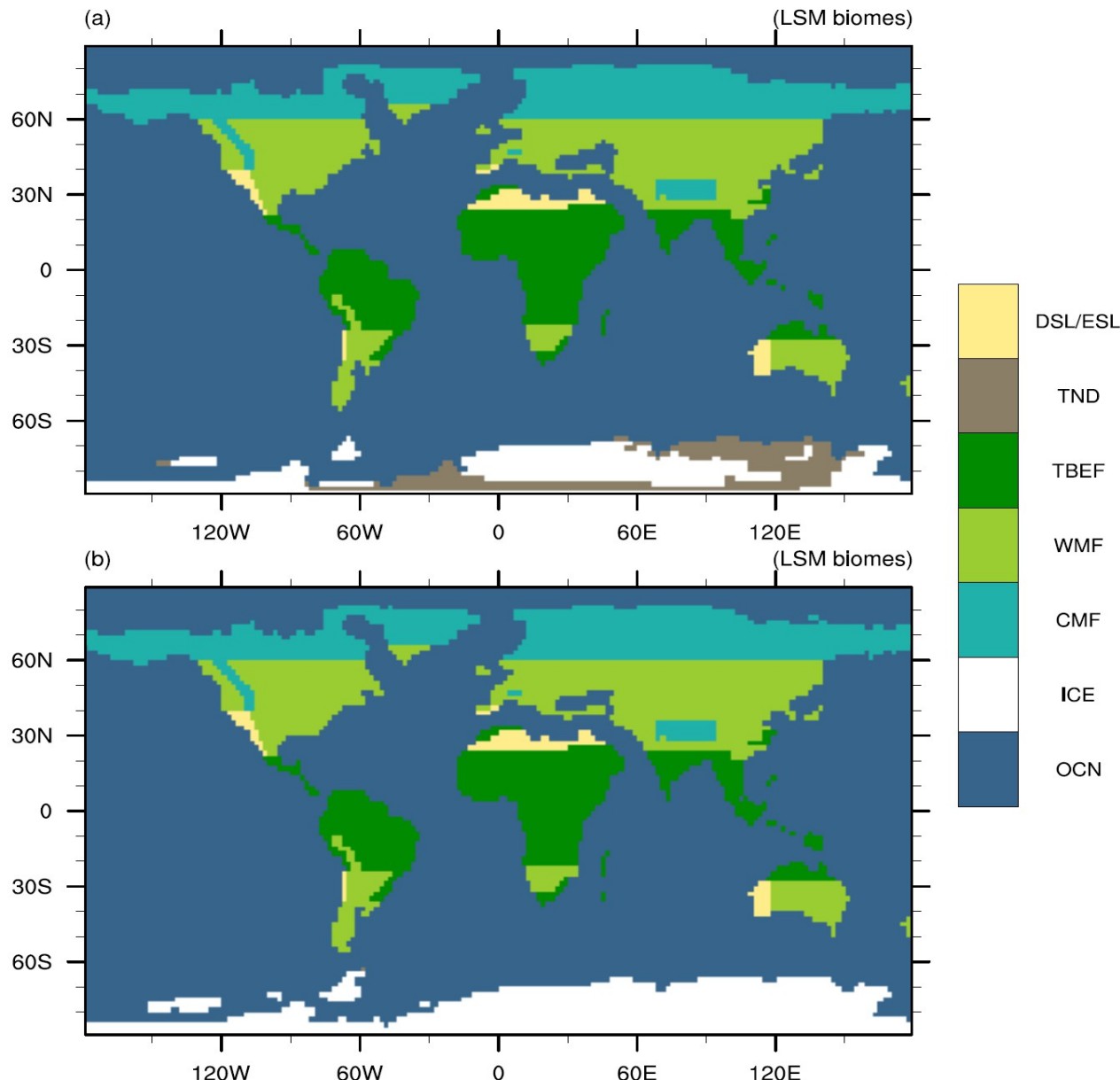

**Figure 5**: **Vegetation reconstruction for the (a) MMCO and the (b) MMG. Colors represent LSM biomes: DSL = deciduous shrubland, ESL = evergreen shrubland, TND = tundra, TBEF = tropical broadleaf evergreen forest, WMF = warm mixed forest, CMF = cool mixed forest, ICE = land ice, OCN = ocean.**

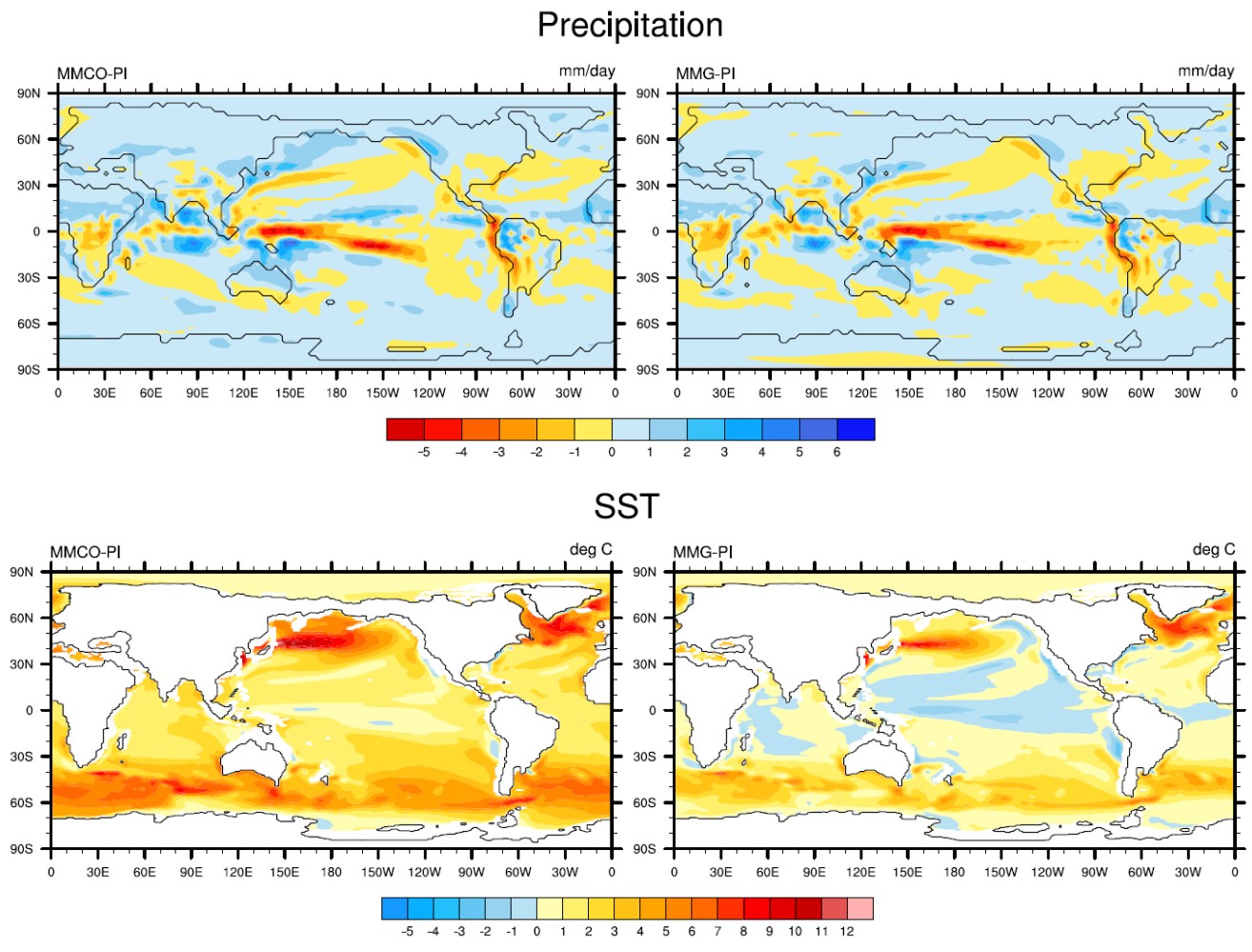

**Figure 6: Sea-surface temperature (SST) (°C) and precipitation (in mm/day) differences between MMCO and MMG experiments, and PI, respectively.**

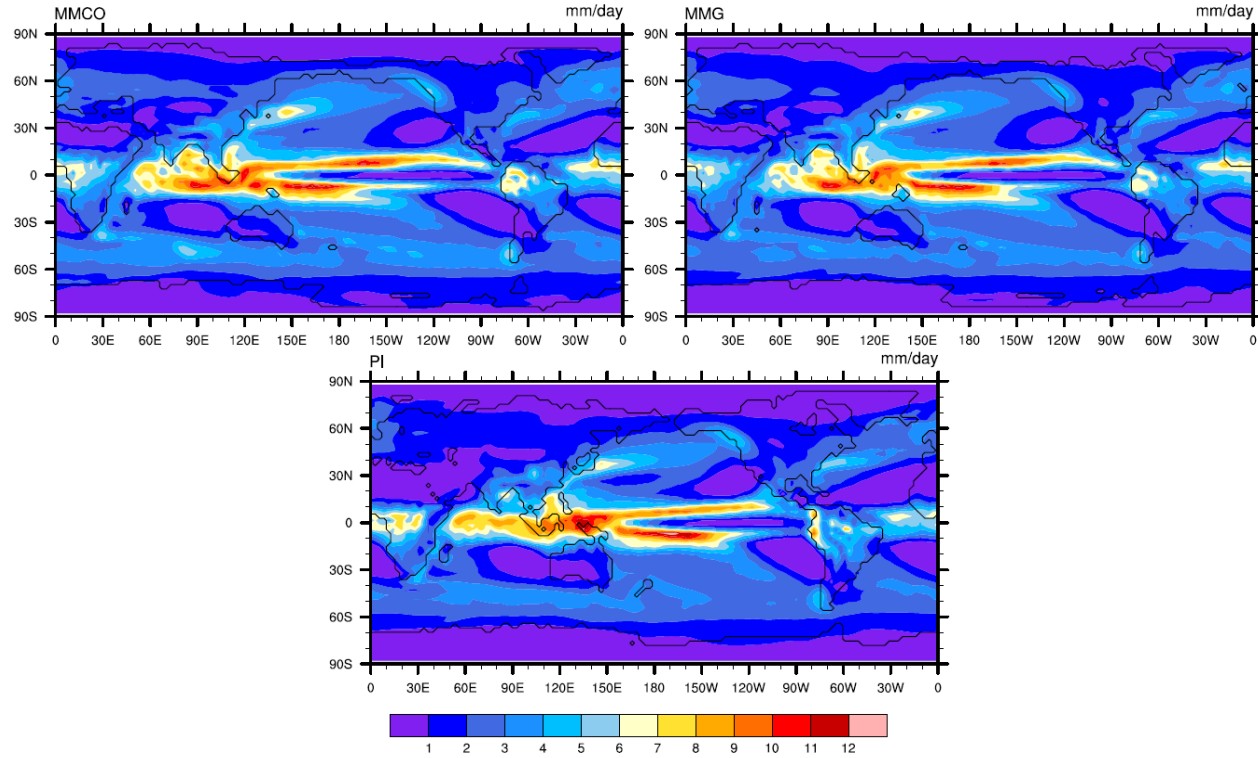

**Figure 7: Precipitation for MMCO, MMG, and PI, in mm/day.**

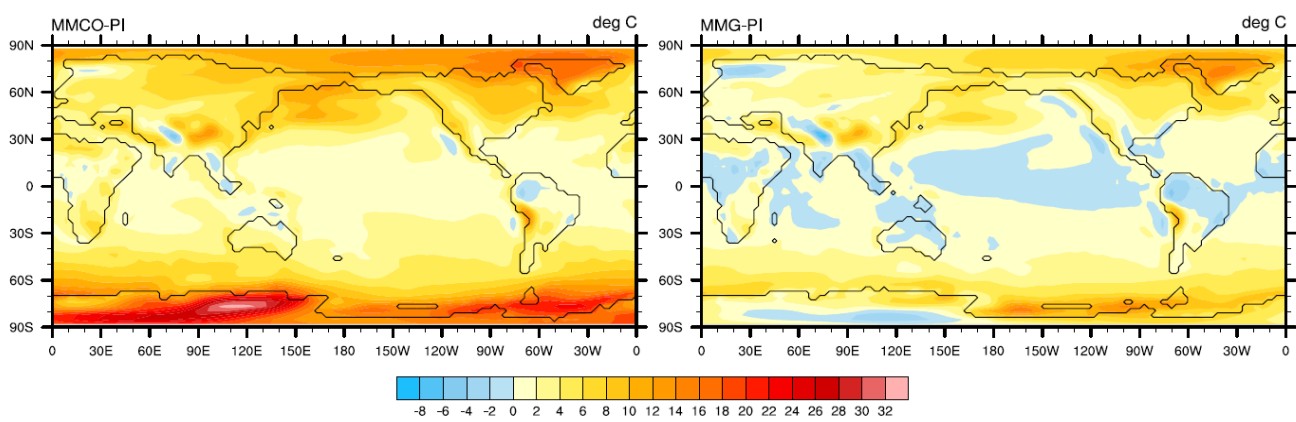

**Figure 8: Surface air temperature (at 2 m height) (°C) differences between MMCO and MMG experiments, and PI, respectively.**

|  | MMCO | MMG |
|---|---|---|
| **Antarctic ice-sheet volume** | 6 million km³ | 23 million km³ |
| **Sea-level** | 48 m higher than at present-day | 5 m higher than at present-day |
| **Atmospheric $CO_2$ concentration** | 400 ppmv | 200 ppmv |
| **Global topography/bathymetry** | mainly Herold et al. (2008) with modifications for tropical seaways (Hall, 2012; Montes et al., 2012) and the Antarctic ice-sheet | same as MMCO, but with global sea-level reduced by 43 m and an expanded Antarctic ice-sheet |
| **Global vegetation** | mainly Pound et al. (2012) with gaps filled according to Wolfe (1985) and Morley (2011); ice and tundra in Antarctica | same as MMCO, but with tundra removed in Antarctica |

**Table 1**: **Summary of boundary conditions for the Middle Miocene Climatic Optimum (MMCO) and the Middle Miocene Glaciation (MMG).**



| BIOME4 | LSM |
|---|---|
| tropical evergreen broadleaf forest | tropical broadleaf evergreen forest |
| tropical deciduous broadleaf forest and woodland | tropical broadleaf deciduous forest |
| warm-temperate evergreen broadleaf and mixed forest | warm mixed forest[1] |
| cool-temperate mixed forest | cool mixed forest[2] |
| tropical savanna | savanna |
| temperate xerophytic shrubland | evergreen shrub land or deciduous shrub land |
| temperate deciduous broadleaf savanna | deciduous shrub land |
| temperate grassland | cool grassland |
| low- and high-shrub tundra | tundra |
| prostrate dwarf-shrub tundra | tundra |
| ice | land ice |
|  |  |
| **Wolfe's (1979) classification** | **LSM** |
| mixed coniferous forest | cool mixed forest[3] |
| tropical rain forest | tropical broadleaf evergreen forest |
| paratropical rain forest | tropical broadleaf evergreen forest[4] |
|  |  |
| **Morley's (2011) classification** | **LSM** |
| megathermal rain forest | tropical broadleaf evergreen forest |
| monsoonal megathermal forest | tropical broadleaf deciduous forest |

**Table 2**: **Conversion of vegetation types to the LSM vegetation scheme.**

[1] The *warm-temperate evergreen broadleaf and mixed forest* may contain broadleaf evergreen trees, needleleaf evergreen trees and deciduous trees. The *warm mixed forest* contains needleleaf evergreen trees and deciduous trees, but not broadleaf evergreen trees. [2] The deciduous trees in the *cool-temperate mixed forest* are temperate, meanwhile the ones in the *cool mixed forest* are boreal. [3] The *mixed coniferous forest* is mainly needleleaf evergreen, but broadleaf trees are also present. These can be deciduous or evergreen. The *cool mixed forest* is formed by needleleaf evergreens and broadleaf deciduous. Broadleaf evergreens are not present. [4] The *paratropical rain forest* is mainly broadleaf evergreen, but it may contain some broadleaf deciduous and conifers.





| Experiment | PI | MMCO | MMG |
|---|---|---|---|
| $CO_2$ | 280 ppmv | 400 ppmv | 200 ppmv |
| $CH_4$ | 760 ppbv | | |
| $N_2O$ | 270 ppbv | | |
| CFC's | 0 | | |
| $O_3$ | 1870 A.D. | | |
| Sulfate aerosols | 1870 A.D. | | |
| Dust and sea salt | PD | same as PI | |
| Carbonaceous aerosols | 30% of PD | | |
| Solar constant | 1365 $Wm^{-2}$ | | |
| Eccentricity | 0.016724 | | |
| Obliquity | 23.446 ° | | |
| Precession | 102.04 ° | | |

**Table 3: Summary of atmospheric composition solar constant, and orbital configuration for the CCSM3 test experiments. PI values are according to Otto-Bliesner et al. (2006). The orbital configuration represents 1950 A.D. values. PD = present day.**
