# Peer review of "Boundary conditions for the Middle Miocene Climate Transition (MMCT v1.0)"

_Geoscientific Model Development, 2017_

## Referee Comment (RC1) · Dr. Langebroek (Referee) · 1 Dec 2017

**Review of Frigola et al., 2017**

This study presents a set of boundary conditions for simulating the climate state before and after the Middle Miocene Climate Transition: the Middle Miocene Climatic Optimum (MMCO) and Middle Miocene Glaciation (MMG), respectively. It also presents an overview of literature on the Antarctic ice sheet configuration, related sea level, atmospheric $CO_2$, paleogeography, topography/bathymetry and vegetation of these time periods.

The manuscript is to a large extent well written and clear. However, certain sections would benefit from more information, while others provide too much detail. I have listed my feedback and suggested changes below.

**GENERAL COMMENTS**

1. More discussion is needed on section 2 Antarctic ice-sheet geometry. There is a nice overview of literature, but no discussion on why the ice sheet configurations of the previous Miocene studies are discarded, and why Pollards Oligocene configuration was used instead. You probably prefer to not use the configurations of Langebroek and Oerlemans, as they use rather simple model configurations. But why do you discard the geometry of Gasson et al.? Related: what forcing and boundary conditions are used in Pollards simulations? How does that compare to the Middle Miocene?

2. Section 4 describing the different published atmospheric $CO_2$ levels is somewhat difficult to follow. A figure showing all the different published records over the Middle Miocene, in combination with horizontal lines indicating your suggestion, would clarify this section. Additionally a discussion on why these values are all so different is needed.

3. Section 5.3, especially lines 204-214 are too detailed. Please make this section more concise. Maybe "We used ArcGIS to convert … to …".

4. Concerning the global topography/bathymetry section: a difference plot to the Herold et al reconstruction (or at least additional information on this) would be highly relevant.

5. Now my biggest concern: The description of the vegetation (Section 6). This section is very lengthily, and to be honest not very useful. In many subsections the vegetation patterns from literature are stated, but then subsequently ignored because you prefer to have a low resolution, simple, distribution. I have no problem with the latter, but I then do not see the use of discussing in detail the vegetation in each continent. I also do see that vegetation might be an important boundary condition, and suggest applying an offline vegetation model (e.g. BIOME4) in order to get a more consistent vegetation pattern within your model set-up. This could then be compared and discussed with previous studies, also previous modelling studies (for example Bradshaw et al., 2012).

6. The final part, the model simulations, are interesting, but need discussion:
   a. How is the grid extended to reach higher southern latitudes? Does this mean that the resolution is lower in the Miocene simulations compared to

the PI simulation? How do you make difference plots then (regridding?)? Does this have an impact on the results?

b. Are the simulations run long enough? What are the trends in the deep ocean (temperatures, salinity, …)?

c. The comparison of the precipitation needs to be rewritten. The lower/higher precipitation along the coast of South America seems to be due to the movement of the continents. Maybe more interesting would be to discuss the apparent shift in the ITCZ. Why?

d. Also the temperature comparison lacks discussion. Why is the MMG simulation warmer than PI? CO2 is lower (200 ppm), right? How different is the Antarctic ice sheet compared to today? Is the cooling in the Pacific caused by changes in gateways/geography/topography? Please discuss.

e. During this discussion please list again the differences between the Miocene simulations (400 vs 200 ppm; different Antarctic ice sheet and vegetation). What is the climate sensitivity of this model? A 200 ppm decrease in CO2 would cause a reduction in temperature of about 2-4°C? Why is there only a difference of 1.6°C? Is the difference larger when you take the global mean surface air temperature? And how much of the cooling is due to the ice expansion (and related albedo changes)? Please discuss.

**SPECIFIC COMMENTS**

1. The start of Section 3 is somewhat confusion, because of the connection between Antarctic ice volume (defined for the Middle Miocene at the end of Section 2) and sea level. Maybe it would be better to start Section 3 with lines 132-136, followed by the discussion of other literature values.

2. Why is the topography over Greenland so high in the Middle Miocene? It looks much higher than a present-day isostatically rebounded topography.

**TECHNICAL COMMENTS**

Line 12: add "successfully" to applied
Lines 20-21: rewrite. $\delta^{18}O$ could also reflect a combined change in ice volume and temperature
Line 25: change "would have been" to "were"
Line 28: explain "important"
Line 40: add "e.g." before references. Using an intermediate complexity climate and ice sheet model, Langebroek et al. (2010) showed that a combination of pCO2 decrease and orbital forcing causes an Antarctic ice sheet expansion that can explain the majority of the benthic $\delta^{18}O$ increase.
Line 54: change "data" to "boundary conditions"
Line 61: change "studies" to "sediment core data"
Line 67: change "simulations" to "study"
Line 93: "This estimate" instead of "This 6 estimate"
Line 104: change "very few" to "little"

Line 190: change "Some" to "Additional". And make clear in this sentence that the modifications will be discussed below.
Line 198: "64" where does this number come from?
Line 443: what does "T42x1" mean? Especially the "x1"?
Line 459: rephrase to " were set to PI following Otto-Bliesner"
Line 471: change "observed" to "simulated"
Line 477: change "complete compilation" to "complete set"
Line 481: change "treated" to "discussed"

Figure 1: caption: change "total elevation" to "surface elevation"
Colours: The colour scale is not great. By colouring 0 to -1000m white, it seems to belong to land, while it is actually ocean. Please change this. Also ice thickness cannot be negative, please update colour bar.

Figure 4: Please make the order of the abbreviations in the caption consistent with the order in the colour bar.

**REFERENCES**
Bradshaw, C. D., Lunt, D. J., Flecker, R., Salzmann, U., Pound, M. J., Haywood, A. M., and Eronen, J. T.: The relative roles of $CO_2$ and palaeogeography in determining late Miocene climate: results from a terrestrial model–data comparison, Clim. Past, 8, 1257-1285, https://doi.org/10.5194/cp-8-1257-2012, 2012.

---

## Referee Comment (RC2) · AJH Henrot (Referee) · 7 Dec 2017

The presented paper describes and provides a set of boundary conditions to be used to force Middle Miocene global climate model simulations. This work focuses on two periods before and after the Middle Miocene Climate Transition (MMCT), the Middle Miocene Climatic Optimum (MMCO) and Middle Miocene Glaciation (MMG). A review of topography, bathymetry, sea-level, Antarctic ice-sheet configuration, atmospheric $CO_2$ concentration, palaeovegetation for the two periods is also presented. The boundary conditions for the MMCO and MMG periods are tested with the Community Climate System Model version 3 (CCSM3). The results of two CCSM3 simulations, for the MMCO and MMG respectively, are briefly presented.

The manuscript is well written and structured. The selection of boundary conditions is described clearly and comprehensively. However, I think that the presentation of the vegetation reconstruction needs to be revised and shortened. The last section presenting the CCSM3 simulation results also lacks discussion, and a comparison with previous General Circulation Model (GCM) simulation results. I therefore recommend this manuscript for publication in GMD, if the authors address the comments listed below.

General Comments

1. Due to the scarcity of palaeovegetation records and the difficulties linked to the identification of plant taxa and correspondence to larger vegetation classes (Plant Functional Types (PFTs) or biomes), the reconstruction of a global vegetation distribution for the Miocene is certainly not easy and subject to many assumptions. Simple and static vegetation maps, mainly based on the reconstruction by Wolfe (1985) have been prescribed in previous modeling studies (Herold et al., 2010; Hamon et al., 2012; Goldner et al., 2014). In that way, deriving a global vegetation map from the reconstruction of Pound et al. (2012), based on the latest palaeovegetation data available, can improve the quality of the vegetation cover to be prescribed. However, the numerous simplifications in the biome classification applied here mask the improvements that could be added to the vegetation reconstruction. The authors end up with a very coarse vegetation distribution, with no differences, except tundra in Antarctica, between the two studied periods, and lack the potential feedback on climate of vegetation change. Wouldn't it be possible to directly interpolate the point-based vegetation reconstruction proposed by Pound et al. (2012) for the Langhian (representative of the MMCO) and for the Serravallian (representative of the MMG) to a 2° map, without so many simplifications, and to keep a maximum of the different biomes listed by Pound et al. (2012)? Corrections could be applied in function of more detailed regional information from Wolfe (1985) and Morley (2011). Then, a translation from BIOME4 to LSM biome classification could be done. However, the number of biome classes should not be too

restricted in order to not loose the distinction between warm/cool and drier biomes, helping to better represent the transition to drier and cooler landscape in the Serravallian (MMG here). If this first option is not possible, I would suggest to extend the number of LSM biomes used here to better represent the vegetation changes between MMCO and MMG. Deriving the vegetation cover from an off-line vegetation model simulation could also be an option to get a global and gridded vegetation map consistent with the model set-up. Previous modeling studies have already done so (Krapp and Junglaus, 2011; Henrot et al., 2017).

2. The last section of the paper, presenting CCSM3 simulations, is too short in comparison to previous sections describing the boundary conditions and lacks a discussion of the simulation results. Evaluating the reliability of the climate simulations would help to prove the suitability of the boundary conditions for Miocene climate modeling. What are the global surface air temperature and precipitation differences between the MMCO, MMG and PI runs? What are the impacts of the boundary conditions changes on the simulated climates? Sensitivity experiments testing separately the impact of boundary condition changes are not presented here, but would it be possible to distinguish or at least discuss the possible impacts of the different boundary condition changes on the simulated climate. The discussion would also benefit from a comparison with previous modeling studies, at least for the MMCO (and even with the same model, see Herold et al. (2010)), and/or with available proxy-data (e.g., for SSTs).

Specific Comments

Introduction: the Introduction would benefit from some description of the climate state of the Middle Miocene, to highlight the differences between MMCO and MMG climate and between the boundary condition sets that will be presented later in the paper.

Lines 36-37: this effect should be taken into account in the vegetation cover reconstruction provided in Section 6.

Lines 43-45: please give the resolution of the boundary conditions and the format they

are available in.

Lines 52-53: the vegetation reconstruction proposed here is not exactly an update of Wolfe (1985). The sentence should be rephrased.

Section 2: a discussion explaining the use of a previous Antarctic topography corresponding to the Oligocene instead of previous published Middle Miocene topographies is needed in Section 2. Some precision could be given concerning the Oligocene configuration and how it is suitable for the Middle Miocene.

Section 4: the presentation of the atmospheric $pCO_2$ estimates is rather confused. A distinction between marine and terrestrial proxy-based reconstructions of atmospheric $pCO_2$ has to be done and discussed. Giving only the $pCO_2$ estimates before and after the MMCT transition (corresponding to the two periods studied, MMCO and MMG) rather than the decrease throughout the transition (lines 155-161) would help to clarify the text. I also suggest adding a graph showing the $pCO_2$ estimates in function of time in Ma. This will help to visualize the uncertainties on $pCO_2$ estimates and the suitability of the two concentrations proposed here for MMCO and MMG.

Line 163-164: 400 ppmv is not a maximum value of $pCO_2$ for the MMCO if you take into account the reconstructions based on stomatal indices (Kürschner et al., 2008), pedogenic carbonates (Retallack, 2009) and recent estimates based on boron isotopes and alkenones (Foster et al., 2012).

Subsection 5.3: the description of the gateway reconstruction is too detailed. I suggest putting lines 204 to 214 to the Appendix.

Section 6:

Line 249: Herold et al. (2010) prescribed a vegetation distribution derived from Wolfe (1985) using a biome classification for CCSM3 adapted from Bonan et al. (2002). Did you use the same classification here? Could you please discuss the eventual differences between the classifications as they are used with the same land-surface

model? I think it could be interesting to add a comparison of your MMCO vegetation reconstruction to the reconstruction proposed in Herold et al. (2010) and to highlight the differences induced by the use of the Pound et al. (2012) dataset.

Line 263-273: I do not agree with the argument proposed here by the authors. The cooling and drying at mid-latitudes has a non-negligible impact on the vegetation distribution (as also stated by the authors in the Introduction, lines 36-37). This effect could be seen on a 2° x 2° resolution map, or even at the T42-resolution used in the CCSM3 simulations with a more detailed biome classification. This vegetation changes can in turn affect the climate-vegetation interactions (even only via the surface albedo changes) and significantly impact on the global climate. I suggest at least revising the vegetation distribution for the MMG and to detail the biome classification used here in order to better represent the changes between MMCO and MMG vegetation distributions (see General Comment 1).

Lines 273-274: how much does the Miocene vegetation distribution differ from the pre-industrial vegetation distribution, as used in CCSM3. It can be useful to briefly list the differences here to better highlight the potential impact of vegetation on the Middle Miocene climate if using the boundary condition set proposed here. I also suggest adding a figure showing the PI vegetation distribution with the same biome classification (maybe in Figure 4).

Subsections 6.1 to 6.9: I suggest making these subsections more concise. I would prefer to have only one paragraph focusing on the major vegetation patterns that are taken here into account for the MMCO and MMG. The detailed description of regional vegetation patterns is useless because most of them are neglected for simplification. The authors can directly refer to Pound et al. (2012) for more detailed information.

Section 7:

Lines 467-475: the presentation and discussion of simulation results need to be reworked and extended. What are the global mean surface air temperature and precipitation differences between the two Miocene runs and the PI run? How do you explain that the MMG run is warmer than the PI run? Is it linked to the absence of ice in the Northern Hemisphere? What is the contribution of the boundary condition changes to the climate differences that the model simulates? A brief comparison with previous modeling studies is highly welcome here. A comparison with some proxy-data (e. g. for SSTs) can also be added.

Concluding remarks: this section needs to be reworked in function of the amendments of the previous sections.

Figures and tables:

Figure 5: I would suggest adding maps of mean surface air temperature differences (MMCO and MMG - PI). It could also be interesting to show the temperature differences between MMCO and MMG.

Table 2: is the correspondence between cool-temperate mixed forest (BIOME 4) and cool mixed forest (LSM) really suitable, since you mention in the footnotes that the cool mixed forest represents only boreal trees? Isn't it another possibility of correspondence?

Table 3: could you please give explicitly the values of the model parameters instead of citing a reference paper? Same for the PI orbital parameters.

Technical comments

Line 50: replace "passages" by "seaways"

Line 51: add the precision "most previous Middle Miocene studies with prescribed vegetation"

Lines 56-57: could you please rephrase this sentence? There are other ways to produce boundary condition assemblages.

Line 93: replace "6 estimate" by "volume estimate"

Line 133: write "previous Section"

Line 165: delete the space between "p" and "CO2"

Line 178: "ice-free conditions"

Line 191: replace "passages" by "seaways"

Line 194: write "Section 2"

Line 230: could you please use "seaway" instead of passage or Central American seaway.

Line 312: "Northeast Australia"

Lines 318, 320: "East Australia"

Line 322 and after: I always put a caption letter to subregions or continents "West Australia", "Southern Africa", etc.

Line 448: please explain configuration T42x1 or detail.

Line 464: "archived as b30.043" does this information really need to be mentioned?

References

Bonan, G.B., Levis, S., Kergoat, L., and Oleson, K.W., 2002. Landscapes as patches of plant functional types: An integrating concept for climate and ecosystem models. Global Biogeochemical Cycles 16, 2.

Foster, G., Lear, C., Rae, J., 2012. The evolution of pCO2, ice volume and climate during the Middle Miocene. Earth Planet. Sci. Lett. 341-344, 243–254.

Goldner, A., Herold, N., Huber, M., 2014. The challenge of simulating the warmth of the mid-Miocene climatic optimum in CESM1. Clim. Past 10, 523–536.

Hamon, N., Sepulchre, P., Donnadieu, Y., Henrot, A.-J., François, L., Jaeger, J.-J., Ramstein, G., 2012. Growth of subtropical forests in Miocene Europe: the roles of

carbon dioxide and Antarctic ice volume. Geology 40, 567–570.

Henrot, A.-J. , Utescher, T., Erdei, B., Dury, M., Hamon, N., Ramstein, G., Krapp, M., Herold, N., Goldner, A., Favre, E., Munhoven, G., François, L., 2017. Middle Miocene climate and vegetation models and their validation with proxy data. Palaeogeography, Palaeoclimatology, Pa- laeoecology 467, 95-119.

Herold, N., Müller, R., Seton, M., 2010. Comparing early to middle Miocene terrestrial climate simulations with geological data. Geosphere 6 (6), 952–961.

Krapp, M., Jungclaus, J., 2011. The Middle Miocene climate as modelled in an atmosphere-ocean-biosphere model. Clim. Past 7, 1169–1188.

Kürschner, W., Kvacek, Z., Dilcher, D., 2008. The impact of Miocene atmospheric carbon dioxide fluctuations on climate and the evolution of terrestrial ecosystems. Proc. Natl. Acad. Sci. 105 (2), 449–453.

Morley, R. J., 2011. Cretaceous and Tertiary climate change and the past distribution of megathermal rainforests, in: Tropical Rainforest Responses to Climatic Change, Bush, M., Flenley, J., Gosling, W., Springer Praxis Books, Berlin, Heidelberg, 1-34.

Pound, M., Haywood, A., Salzmann, U., Riding, J., 2012. Global vegetation dynamics and latitudinal temperature gradients during the Mid to Late Miocene (15.97–5.33 Ma). Earth Sci. Rev. 112 (42), 1–22.

Retallack, G., 2009. Refining a pedogenic-carbonate CO2 paleobarometer to quantify a middle Miocene greenhouse spike. Palaeogeogr. Palaeoclimatol. Palaeoecol. 281 (2), 57–65.

Wolfe, J., 1985. Distribution of major vegetational types during the Tertiary. Geophys. Monogr. 32, 357–375.

---

## Author Comment (AC1) · 5 Mar 2018

Dear Dr. Langebroek,

Thanks for the constructive comments. The manuscript has been clarified and improved by taking your comments into account.

Notes: unless otherwise specified, line numbers refer to the non-updated manuscript version. The authors' comments are in blue, and the changes in the manuscript are in green.

**General Comments**

1. More discussion is needed on section 2 Antarctic ice-sheet geometry. There is a nice overview of literature, but no discussion on why the ice sheet configurations of the previous Miocene studies are discarded, and why Pollard's Oligocene configuration was used instead. You probably prefer to not use the configurations of Langebroek and Oerlemans, as they use rather simple model configurations. But why do you discard the geometry of Gasson et al.? Related: what forcing and boundary conditions are used in Pollard's simulations? How does that compare to the Middle Miocene?

As the reviewer mentions, the reason why Oerlemans and Langebroek's configurations were discarded is that they use rather simple model configurations. We wanted to provide a characterization of the Antarctic ice-sheet in two dimensions, i.e. varying with both latitude and longitude, and this is not available from Oerlemans or Langebroek's studies.

The scope of the current study was to provide Antarctic topography data consistent with published Antarctic ice volume estimates for the MMCO and MMG, and this was successfully accomplished. The configuration of Gasson et al. (2016) could be considered in future sensitivity studies, because uncertainties in the ice-volume estimates are high, but we consider the data from Pollard's simulations used here definitely suitable for the Middle Miocene since there is little to link those data to a specific time period (except for the Laskar orbits) (see below). Additionally, the distribution of ice in our study is comparable to that in Gasson et al. (2016): for the MMG, a continental scale ice-sheet exists in East Antarctica with ice thicknesses of ~3000-4000 m, although in West Antarctica there is less ice in Pollard's data; for the MMCO, the ice-sheets occupy similar positions, although they are less extensive in Pollard's data.

Regarding David Pollard's simulations, the physical model used is close to that described in Pollard and DeConto (2012), but with no marine ice physics, so that any floating ice is immediately removed. The bedrock-elevation boundary conditions are from the modern ALBMAPv1 dataset (Le

Brocq et al., 2010).

Climate forcing is obtained from a matrix of previous Global Climate Model (GCM) climates for various orbits, CO2 levels and ice sizes. The GCM used is GENESIS version 3 (as in Alder et al., 2011, except with a slab mixed layer ocean). Three Earth orbits are used, with eccentricity, precession and obliquity set corresponding to warm, intermediate and cold austral summers. Three CO2 levels are used, spanning the range in the long term run. Three Antarctic ice sizes (continental, ~half and no ice cover) are specified. 10-year mean equilibrated GCM climate solutions (i.e., monthly mean surface air temperatures and precipitation) are saved for all combinations of orbit, CO2 and ice size, yielding a matrix of 27 climates.

In the long-term ice-sheet run, the appropriate climate at any point in the run is obtained by linearly weighting the surrounding saved climates in the matrix, with the weights proportional to the current austral summer insolation, ice size and logarithm of CO2 level. This matrix-forcing method is discussed further in Pollard (2010). The annual surface mass balance at each point on the ice sheet is calculated from the monthly surface air temperatures and precipitation, using a simple box (zero-dimensional) seasonal surface-mass model that includes snow storage and refreezing of meltwater, and surface melting based on Positive Degree Days (Pollard and DeConto, 2012). The run is initialized with (essentially) no ice. Insolation is based on Laskar et al. (2004). The run is 12 Myr long, nominally from "37 Ma to 25 Ma", although there is little to actually link it to specific paleodates except the Laskar orbits. From 37 to 33 Ma, CO2 decreases linearly from 6xPAL to 2xPAL. From 33 to 25 Ma, CO2 increases linearly from 2xPAL to 10xPAL (where PAL= 280 ppmv). The configuration used to represent MMCO conditions corresponds to 34.8 Ma (CO2 = 3.8xPAL). The one representing MMG conditions, to 33 Ma (CO2 = 2xPAL).

Lines 82-99 (old numeration) have been rewritten.

2. Section 4 describing the different published atmospheric CO2 levels is somewhat difficult to follow. A figure showing all the different published records over the Middle Miocene, in combination with horizontal lines indicating your suggestion, would clarify this section. Additionally a discussion on why these values are all so different is needed.

The reviewer's suggestion is interesting, although we do not see the addition of a figure as a requirement for the comprehension of the CO2 section. Nevertheless, lines 162-168 (now lines 205-214) have been rephrased as follows to make the section more clear:

'We chose atmospheric $CO_2$ concentrations of 400 ppmv and 200 ppmv to represent the MMCO and

the MMG respectively (Table 1). Although somewhat arbitrary, these values are within the range of published estimates. The 400 ppmv MMCO is in favourable agreement with Foster et al. (2012) (~392 ppmv at ~15.8 Ma) and Tripati et al. (2009) (~430 ppmv at ~15.1 Ma), although higher than Pearson et al. (2000) (~300 ppmv at ~16.2 Ma) and Pagani et al. (2005) (~300 ppmv at ~15 Ma), and lower than Kürschner et al. (2008) (> ~400-500 ppmv at ~15.5 Ma) and Retallack (2009) (~852 ppmv at ~15.6 Ma) maxima. The 400 ppmv estimate is also in good agreement with the most recent alkenone- and boron isotope-based pCO2 reconstructions for the MMCO by Zhang et al. (2013) and Greenop et al. (2014). The 200 ppmv MMG estimate is in good agreement with Foster et al. (2012) (~200 ppmv at ~12 Ma) and Pagani et al. (2005) (~200 ppmv at ~13 Ma), although higher than Pearson et al. (2000) (~140 ppmv at ~14.7 Ma) and Retallack (2009) (~116 ppmv at ~14.6 Ma), and lower than Tripati et al. (2009) (~340 ppmv at ~12 Ma) and Kürschner et al. (2008) (~280 ppmv at ~14 Ma) minima.'

The difference in the $CO_2$ estimates between the various studies arises most likely from method-related uncertainties and/or the relatively coarse temporal resolution of some of the datasets.

We added a note at line 185 (new numeration):
'The difference in the $CO_2$ estimates between the various studies arises most likely from method-related uncertainties and/or the relatively coarse temporal resolution of some of the datasets'.

3. Section 5.3, especially lines 204-214 are too detailed. Please make this section more concise. Maybe "We used ArcGIS to convert ... to ..."

The text at lines 204-214 (old numeration) has been shortened:
'South East Asian paleogeography was modified based on Hall's (2012) reconstruction constrained at 15 Ma (Fig. 3). Hall's data, available as a georeferenced image, were converted into grid format using ArcGIS. Qualitative height/depth values were assigned to the different geographic features: ~2800 m for volcanoes, ~1000 m for highlands, ~250 m for land, ~-22 m for carbonate platforms, ~-200 m for shallow sea, <-4000 m for deep sea, and ~-5500 m for trenches. After embedding the data into the MMCO global dataset, minor manual smoothing was applied at the margins of the embedded region. Here, shallow bays were removed and single, shallow grid points surrounded by much deeper grid points were deepened to the adjacent depth. In total, these modifications affected ~0.5% of the total number of grid points.'

4. Concerning the global topography/bathymetry section: a difference plot to the Herold et al reconstruction (or at least additional information on this) would be highly relevant.

A new figure has been added (and a reference to it at line 243 (new numeration)).

[Figure]

**Figure S1: Difference between MMCO and the topography/bathymetry by Herold et al. (2008), in meters. Sea level is 4 m higher in the MMCO dataset (subsection 5.2), the Indonesian Throughflow barriers are shallower (subsection 5.3), the Panama seaway is narrower (subsection 5.4), and the Antarctic topography/bathymetry is based on David Pollard's data (subsection 5.1) and consistent with MMCO ice volume estimates.**

5. Now my biggest concern: The description of the vegetation (Section 6). This section is very lengthily, and to be honest not very useful. In many subsections the vegetation patterns from literature are stated, but then subsequently ignored because you prefer to have a low resolution, simple, distribution. I have no problem with the latter, but I then do not see the use of discussing in detail the vegetation in each continent. I also do see that vegetation might be an important boundary condition, and suggest applying an offline vegetation model (e.g. BIOME4) in order to get a more consistent vegetation pattern within your model set-up. This could then be compared and discussed with previous studies, also previous modelling studies (for example Bradshaw et al., 2012).

Although a detailed discussion on the vegetation of each region is not indispensable for the comprehension of this manuscript, we think that it is important to show how exactly we decided what vegetation to assign to each region.

Subsections 6.1-6.9 have thus been moved to the Appendix (Sect. 10), in case the reader was interested in those details. A reference to the Appendix has been added at line 360 (new numeration).

The reviewer's suggestion of using the output from an offline vegetation model as a boundary condition is very interesting (e.g., the ones described in Henrot et al., 2017), although here our aim was to provide boundary conditions based on palaeobotanical data. The vegetation output from an offline vegetation model is based on a climatic forcing. In our study the approach was the opposite: using vegetation data to be able to produce a climatic output. An alternative for GCMs including a dynamic vegetation component would be to use our Middle Miocene vegetation dataset to initialize the vegetation model. Nevertheless, we consider that, although coarse, our dataset provides a fair characterization of Middle Miocene global vegetation patterns.

A note has been added at line 355 (new numeration).

6. The final part, the model simulations, are interesting, but need discussion:

a. How is the grid extended to reach higher southern latitudes? Does this mean that the resolution is lower in the Miocene simulations compared to the PI simulation? How do you make difference plots then (regridding?)? Does this have an impact on the results?

The Miocene grid is a dipole grid created from scratch using the CCSM3 setup tools described in Rosenbloom et al. (2011) and defined by the following parameters: dyeq=0.25 (meridional grid spacing at the equator, in degrees), dsig=20 (Gaussian e-folding scale at equator), and jcon=45 (rows of constant meridional grid spacing at poles).
In some areas the Miocene grid presents a slightly coarser resolution than the PI grid, since both grids have the same number of grid points (384x320) and the Miocene grid reaches further south than the PI grid (~87°S vs ~79°S).
Difference plots are made by regridding from the PI grid onto the Miocene grid. The method used is the "patch recovery" method (http://www.ncl.ucar.edu/Applications/ESMF.shtml), which gives better approximations than the "bilinear" method. We do not think interpolation has any significant effect on the results.

The reviewer's comment has been addressed in the manuscript at lines 549-556 (new numeration).

b. Are the simulations run long enough? What are the trends in the deep ocean (temperatures, salinity, …)?

The temperature trends in the deep ocean (at 4-5 km depth) are < 0.14, 0.15, and 0.17 °C/100 years in the PI, MMCO, and MMG cases, respectively. At that same depth range, the salinity trends are <0.01, 0.007, and 0.01 psu/100 years for PI, MMCO, and MMG, respectively. These values represent quasi-equilibrium conditions and we consider them sufficiently small for the focus of this study.

The reviewer's comment has been addressed in the manuscript at lines 565-568 (new numeration).

c. The comparison of the precipitation needs to be rewritten. The lower/higher precipitation along the coast of South America seems to be due to the movement of the continents. Maybe more interesting would be to discuss the apparent shift in the ITCZ. Why?

We checked again the absolute precipitation maps (see Figure S2) and the Miocene experiments present lower precipitation rates than PI along the northwest coast of South America. Nevertheless, the difference of 5-6 mm/day we suggested is too high, and as the reviewer noted, linked to the movement of the continents.

The text has been modified by replacing 'up to 5-6 mm/day lower' with '3-4 mm/day lower' at line 572 (new numeration). We also added a paragraph on the ITCZ. Please, check lines 575-579 (new numeration) for more details. Additionally, a new figure (Fig. S2) has been added (please, see below).

[Figure]

**Figure S2: Precipitation for MMCO, MMG, and PI, in mm/day.**

d. Also the temperature comparison lacks discussion. Why is the MMG simulation warmer than PI? CO2 is lower (200 ppm), right? How different is the Antarctic ice sheet compared to today? Is the cooling in the Pacific caused by changes in gateways/geography/topography? Please discuss.

Indeed, CO2 is lower in the MMG simulation than in the control run (MMG: CO2 = 200 ppmv, PI: CO2 = 280 ppmv). Nevertheless, SST's are higher for MMG (18.04°C) than for PI (16.85°C). Potential causes for a MMG climate warmer than PI could be the lower extent of ice-sheets (the Antarctic ice-sheet is smaller and the northern Hemisphere free of ice-sheets in the MMG run), or the different vegetation cover (Knorr et al., 2011). However, unambiguously disentangling the effects of each of the different boundary conditions would require performing a series of sensitivity experiments, which was beyond the scope of the current study. Here our aim was testing the idoneity of the current boundary conditions as input data in GCMs for MMCO and MMG experiments.

In the MMG experiment the Antarctic ice-sheet has a volume of 23 million km³, hence lower than present-day (27 million km³, according to Fretwell et al., 2013).

This point has been addressed at lines 592-598 (new numeration).

e. During this discussion please list again the differences between the Miocene simulations (400 vs 200 ppm; different Antarctic ice sheet and vegetation). What is the climate sensitivity of this model? A 200 ppm decrease in CO2 would cause a reduction in temperature of about 2-4°C? Why is there only a difference of 1.6°C? Is the difference larger when you take the global mean surface air temperature? And how much of the cooling is due to the ice expansion (and related albedo changes)? Please discuss.

The climate sensitivity of CCSM3 is discussed in Kiehl et al. (2006), where two different approaches are used, one based on results from a slab ocean run with fixed CO2 and the other one based on a fully coupled run with increasing CO2 rates. The results obtained are 2.47°C and 1.48°C, respectively.

When, instead of SSTs, surface air temperatures (at 2 m height) are considered, our results show mean global values of 16.38°C, 13.88°C, and 12.16°C for the MMCO, MMG, and PI, respectively. This implies a decrease of 2.5°C between the MMCO (CO2= 400 ppmv + small Antarctic ice-sheet) and MMG (CO2= 200 ppmv + expanded ice-sheet), which is in good agreement with the CCSM3 climate sensitivity values suggested in Kiehl et al. (2006). A decrease of 1.6°C in SST's would also be in agreement with Kiehl et al. (2006).

Quantifying how much of the cooling is due to ice expansion is a very interesting suggestion, although it would require performing a series of sensitivity studies, with fixed CO2 and varying Antarctice ice volume, which were beyond the scope of the current study. Here our aim was testing the idoneity of the current boundary conditions as input data in GCMs for MMCO and MMG experiments.

The reviewer's comment has been addressed at lines 584-588 (new numeration).

**Specific Comments**

1. The start of Section 3 is somewhat confusion, because of the connection between Antarctic ice volume (defined for the Middle Miocene at the end of Section 2) and sea level. Maybe it would be better to start Section 3 with lines 132-136, followed by the discussion of other literature values.

Lines 132-136 have been moved to the top of the section. However, those lines have been slightly rephrased because they contained a reference to Equation (1), which had not been defined yet.

Lines 112-118 were removed because they had become redundant.

2. Why is the topography over Greenland so high in the Middle Miocene? It looks much higher than a present-day isostatically rebounded topography.

Our values are based on Herold et al. (2008). In that study, the topography over Greenland is "reduced by 2300 m" compared to present-day and "isostatically corrected by 1651 m", which means that it is 649 m lower than at present-day. We compared our topography to Bamber et al. (2001) present-day isostatically rebounded topography (Figure 5 in Bamber et al., 2001). We agree with the reviewer that our topography is a bit higher, reaching maximum values of ~2400 m, versus maximum values of ~2000 m in Bamber et al. (2001). Nevertheless, we believe that Herold's values are still a good approximation of an ice-free Greenland topography.

**Technical Comments**

-Line 12: add "successfully" to applied.

Added.

-Lines 20-21: rewrite. δ18O could also reflect a combined change in ice volume and temperature.

We added this text:
'or a combination of both'.

-Line 25: change "would have been" to "were".

Changed.

-Line 28: explain "important".

A reference to Section 3 has been added. In this section, sea level fall published estimates for the Middle Miocene Climate Transition are reviewed in detail.

-Line 40: add "e.g." before references. Using an intermediate complexity climate and ice sheet model, Langebroek et al. (2010) showed that a combination of pCO2 decrease and orbital forcing

causes an Antarctic ice sheet expansion that can explain the majority of the benthic δ18O increase.

Added at line 47(new numeration), although slightly rephrased:
'Langebroek et al. (2010), for example, using an isotope enabled ice-sheet–climate model forced with a pCO2 decrease and varying time-dependent orbital parameters, modeled an increase in δ18O of sea water in good agreement with published MMCT estimates.'

-Line 54: change "data" to "boundary conditions".

Changed.

-Line 61: change "studies" to "sediment core data".

'Studies' has been replaced with 'sediment core studies'.

-Line 67: change "simulations" to "study".

Changed (now line 77).

-Line 93: "This estimate" instead of "This 6 estimate".

Done (now line 95).

-Line 104: change "very few" to "little".

Changed.

-Line 190: change "Some" to "Additional". And make clear in this sentence that the modifications will be discussed below.

Done.

A comment has been added at line 191 (now line 243):
'(see discussion below in subsections 5.1–5.4)'.

-Line 198: "64" where does this number come from?

The 64 m present-day sea-level equivalent value is in good agreement with Vaughan et al. (2013) (58.3 m for the Antarctic ice-sheet and 7.36 m for the Greenland ice-sheet).

The following text has been added at line 251 (new numeration):
'This present-day estimate is in good agreement with Vaughan et al. (2013) (58.3 m for the Antarctic ice-sheet and 7.36 m for the Greenland ice-sheet).'

-Line 448: what does "T42x1" mean? Especially the "x1"?

T42 is the atmosphere horizontal grid, a Gaussian grid with 64 points in latitude and 128 points in longitude (~2.8° resolution). The notation T42 refers to the spectral truncation level. x1 is the ocean horizontal grid, a dipole grid with 384 points in latitude and 320 points in longitude. The zonal resolution of the ocean horizontal grid is ~1°, the mean meridional resolution is ~0.5°, refined around the equator (~0.3°). The notation x1 refers to the nominal zonal resolution. T42x1 is the model configuration employing the T42 and x1 grids.

Lines 448-453 (old numeration) have been modified as follows:
'The atmosphere horizontal grid employed in the PI run, T42, is a Gaussian grid with 64 points in latitude and 128 points in longitude (~2.8° resolution). The notation T42 refers to the spectral truncation level. The land and atmosphere models share the same horizontal grid. The ocean horizontal grid, x1, is a dipole grid with 384 points in latitude and 320 points in longitude. The zonal resolution of the ocean horizontal grid is ~1°, the mean meridional resolution is ~0.5°, refined around the equator (~0.3°). The notation x1 refers to the nominal zonal resolution. The ocean and sea–ice components share the same horizontal grid. The atmosphere and ocean vertical grids have 26 and 40 vertical levels, respectively. This model grid configuration is known as T42x1.'

-Line 459: rephrase to "were set to PI following Otto-Bliesner".

Rephrased.

-Line 471: change "observed" to "simulated".

The word "observed" does not appear in the text anymore. We rewrote that part of the text in

-Line 477: change "complete compilation" to "complete set".

Changed.

-Line 481: change "treated" to "discussed".

Changed.

-Figure 1: caption: change "total elevation" to surface elevation". Colours: The colour scale is not great. By colouring 0 to -1000 white, it seems to belong to land, while it is actually ocean. Please change this. Also ice thickness cannot be negative, please update colour bar.

Done.

-Figure 4: Please make the order of the abbreviations in the caption consistent with the order in the colour bar.

Done.

**Additional modifications:**

-Line 73 (now line 83):
In Langebroek et al. (2010) the model is isotope-enabled, but in Langebroek et al. (2009) it is not.

We have thus rephased 'Langebroek et al. (2009) used a coupled isotope-enabled ice-sheet–climate model' as 'Langebroek et al. (2009) used a coupled ice-sheet–climate model'.

We hope we have addressed all your comments.

Yours sincerely,

Amanda Frigola and co-authors.

**References:**

Alder, J. R., Hostetler, S. W., Pollard, D., and Schmittner, A.: Evaluation of a present-day climate simulation with a new coupled atmosphere-ocean model GENMOM, GMD, 4(1), 69–83, doi:10.5194/gmd-4-69-2011, 2011.

Bamber, J. L., Layberry, R. L., and Gogineni, S. P.: A new ice thickness and bed data set for the Greenland ice sheet 1. Measurement, data reduction, and errors, J. Geophys. Res.-Atmos., 106(D24), 33773-33780, doi:10.1029/2001JD900054, 2001.

Fretwell, P., Pritchard, H. D., Vaughan, D. G., Bamber, J. L., Barrand, N. E., Bell, R., Bianchi, C., et al.: Bedmap2: improved ice bed, surface and thickness datasets for Antarctica, Cryosphere, 7, 375–393, doi:10.5194/tc-7-375-2013, 2013.

Gasson, E., DeConto, R. M., Pollard, D. and Levy, R. H.: Dynamic Antarctic ice sheet during the early to mid-Miocene, P. Natl. Acad. Sci., 113(13), 3459–3464, doi:10.1073/pnas.1516130113, 2016.

Groeneveld, J., Henderiks, J., Renema, W., McHugh, C. M., De Vleeschouwer, D., Christensen, B. A., Fulthorpe, C. S., Reuning, L., Gallagher, S. J., Bogus, K., Auer, G., Ishiwa, T., and Expedition 356 Scientists: Australian shelf sediments reveal shifts in Miocene Southern Hemisphere westerlies, Sci. Adv., 3(5), 1–9, doi:10.1126/sciadv.1602567, 2017.

Henrot, A.-J., Utescher, T., Erdei, B., Dury, M., Hamon, N., Ramstein, G., Krapp, M., Herold, N., Goldner, A., Favre, E., Munhoven, G., and François, L.: Middle Miocene climate and vegetation models and their validation with proxy data, Palaeogeogr. Palaeoclimatol. Palaeoecol., 467, 95–119, doi:10.1016/j.palaeo.2016.05.026, 2017.

Herold, N., Seton, M., Müller, R. D., You, Y. and Huber, M.: Middle Miocene tectonic boundary conditions for use in climate models, Geochem. Geophy. Geosy., 9(10), Q10009, doi:10.1029/2008GC002046, 2008.

Holbourn, A., Kuhnt, W., Regenberg, M., Schulz, M., Mix, A., and Andersen, N.: Does Antarctic glaciation force migration of the tropical rain belt?, Geology, 38(9), 783–786, doi:10.1130/G31043.1, 2010.

Kiehl, J. T., Shields, C. A., Hack, J. J., Collins, W. D.: The Climate Sensitivity of the Community Climate System Model Version 3 (CCSM3), J. Climate, 19, 2584–2596, doi:10.1175/JCLI3747.1, 2006.

Knorr, G., Butzin, M., Micheels, A., and Lohmann, G.: A warm Miocene climate at low

atmospheric CO2 levels, Geophys. Res. Lett., 38, L20701, 1–5, doi:10.1029/2011GL048873, 2011.

Langebroek, P. M., Paul, A. and Schulz, M.: Antarctic ice-sheet response to atmospheric CO2 and insolation in the Middle Miocene, Clim. Past, 5(4), 633–646, doi:10.5194/cp-5-633-2009, 2009.

Langebroek, P. M., Paul, A. and Schulz, M.: Simulating the sea level imprint on marine oxygen isotope records during the middle Miocene using an ice sheet-climate model, Paleoceanography, 25(4), PA4203, doi:10.1029/2008PA001704, 2010.

Laskar, J., Robutel, P., Joutel, F., Gastineau, M., Correia, A. C. M., Levrard, B.: A long term numerical solution for the insolation quantities of the Earth, 428, 261-285, A&A, doi:10.1051/0004-6361:20041335, 2004.

Le Brocq, A. M., Payne, A. J., and Vieli, A.: An improved Antarctic dataset for high resolution numerical ice sheet models (ALBMAP v1), Earth Syst. Sci. Data., 2 (2), 247-260, doi:10.5194/essd-2-247-2010, 2010.

Pollard, D.: A retrospective look at coupled ice sheet–climate modeling, Clim. Change, 100, 173-194, doi:10.1007/s10584-010-9830-9, 2010.

Pollard, D. and DeConto, R. M.: Description of a hybrid ice sheet-shelf model, and application to Antarctica, Geosci. Model Dev., 5, 1273-1295, doi:10.5194/gmd-5-1273-2012, 2012.

Rosenbloom, N., Shields, C., Brady, E., Levis, S. and Yeager, S.: Using CCSM3 for Paleoclimate Applications, NCAR Technical Note NCAR/TN–483+STR, National Center for Atmospheric Research, Boulder, Colorado, 1–81, doi:10.5065/D69S1P09, 2011.

Vaughan, D. G., Comiso, J. C., Allison, I., Carrasco, J., Kaser, G., Kwok, R., Mote, P., Murray, T., Paul, F., Ren, J., Rignot, E., Solomina, O., Steffen, K., and Zhang, T.: Observations: Cryosphere, in: Climate Change 2013: The Physical Science Basis. Contribution of Working Group I to the Fifth Assessment Report of the Intergovernmental Panel on Climate Change, edited by Stocker, T. F., Qin, D., Plattner, G. K., Tignor, M., Allen, S. K., Boschung, J., Nauels, A., Xia, Y., Bex, V., and Midgley, P. M., Cambridge University Press, Cambridge, United Kingdom and New York, NY, USA, 317-382, 2013.

---

## Author Comment (AC2) · 5 Mar 2018

Dear Dr. Henrot,

Thanks for the constructive comments. The manuscript has been clarified and improved by taking your comments into account.

Notes: unless otherwise specified, line numbers refer to the non-updated manuscript version. The authors' comments are in blue, and the changes in the manuscript are in green.

**General comments:**

1. Due to the scarcity of palaeovegetation records and the difficulties linked to the identification of plant taxa and correspondence to larger vegetation classes (Plant Functional Types (PFTs) or biomes), the reconstruction of a global vegetation distribution for the Miocene is certainly not easy and subject to many assumptions. Simple and static vegetation maps, mainly based on the reconstruction by Wolfe (1985) have been prescribed in previous modeling studies (Herold et al., 2010; Hamon et al., 2012; Goldner et al., 2014). In that way, deriving a global vegetation map from the reconstruction of Pound et al. (2012), based on the latest palaeovegetation data available, can improve the quality of the vegetation cover to be prescribed. However, the numerous simplifications in the biome classification applied here mask the improvements that could be added to the vegetation reconstruction. The authors end up with a very coarse vegetation distribution, with no differences, except tundra in Antarctica, between the two studied periods, and lack the potential feedback on climate of vegetation change. Wouldn't it be possible to directly interpolate the point-based vegetation reconstruction proposed by Pound et al. (2012) for the Langhian (representative of the MMCO) and for the Serravallian (representative of the MMG) to a 2° map, without so many simplifications, and to keep a maximum of the different biomes listed by Pound et al. (2012)?

Pound et al. (2012) dataset definitely constitutes an improvement in terms of the characterization of the Middle Miocene vegetation patterns. Nevertheless interpolation is problematic since some vast areas (e.g. Africa) present still low data coverage. Extrapolation is thus required. The difficulty arises in how to appropriately extrapolate the palaeobotanical information from isolated data points into the area surrounding them, i.e, in how to decide what exact area around those data points can be represented by that same vegetation pattern. Plus extrapolation of biomes is conceptually complicated: what is for example the appropriate biome that should be assigned to a grid point close to a "deciduous shrub land" and a "tropical broadleaf evergreen forest"? Wolfe and Morley performed that extrapolation in their studies, but Pound et al. (2012) did not. That is the reason why

we often opted for using Wolfe and Morley's data in regions where Pound et al.'s dataset presented low coverage.

Corrections could be applied in function of more detailed regional information from Wolfe (1985) and Morley (2011). Then, a translation from BIOME4 to LSM biome classification could be done. However, the number of biome classes should not be too restricted in order to not loose the distinction between warm/cool and drier biomes, helping to better represent the transition to drier and cooler landscape in the Serravallian (MMG here). If this first option is not possible, I would suggest to extend the number of LSM biomes used here to better represent the vegetation changes between MMCO and MMG.

Please, see below our reply to the reviewer's comments to lines 263-273.

Deriving the vegetation cover from an off-line vegetation model simulation could also be an option to get a global and gridded vegetation map consistent with the model set-up. Previous modeling studies have already done so (Krapp and Junglaus, 2011; Henrot et al., 2017).

The reviewer's suggestion of using the output from an offline vegetation model as a boundary condition is very interesting (e.g., the ones described in Henrot et al., 2017), although here our aim was to provide boundary conditions based on palaeobotanical data. The vegetation output from an offline vegetation model is based on a climatic forcing. In our study the approach was the opposite: using vegetation data to be able to produce a climatic output. An alternative for GCMs including a dynamic vegetation component would be to use our Middle Miocene vegetation dataset to initialize the vegetation model. Nevertheless, we consider that, although coarse, our dataset provides a fair characterization of Middle Miocene global vegetation patterns.

A note has been added at line 355 (new numeration).

2. The last section of the paper, presenting CCSM3 simulations, is too short in comparison to previous sections describing the boundary conditions and lacks a discussion of the simulation results. Evaluating the reliability of the climate simulations would help to prove the suitability of the boundary conditions for Miocene climate modeling. What are the global surface air temperature and precipitation differences between the MMCO, MMG and PI runs? What are the impacts of the boundary conditions changes on the simulated climates? Sensitivity experiments testing separately

the impact of boundary condition changes are not presented here, but would it be possible to distinguish or at least discuss the possible impacts of the different boundary condition changes on the simulated climate. The discussion would also benefit from a comparison with previous modeling studies, at least for the MMCO (and even with the same model, see Herold et al. (2010)), and/or with available proxy-data (e.g., for SSTs).

Please, see below our reply to the reviewer's comments to Section 7. However, a detailed model-model or even model-data comparison is beyond the scope of this paper and will be the subject of future studies.

**Specific comments:**

Introduction: the Introduction would benefit from some description of the climate state of the Middle Miocene, to highlight the differences between MMCO and MMG climate and between the boundary condition sets that will be presented later in the paper.

We have added a short description of the Middle Miocene climate and we have linked it to the Middle Miocene Climate Transition.

Lines 17-18 (now lines 17-21) have been rewritten:
'The Middle Miocene (ca. 16–11.6 Ma) was marked by important changes in global climate. The first stage of this time period, the Middle Miocene Climatic Optimum (MMCO), was characterized by warm conditions, comparable to those of the late Oligocene. Although global climate remained warmer than present-day during the whole Miocene (Pound et al., 2012), an important climate transition associated with major Antarctic ice-sheet expansion and global cooling took place between ~15 and 13 Ma, the so called Middle Miocene Climate Transition (MMCT).'

Lines 36-37: this effect should be taken into account in the vegetation cover reconstruction provided in Section 6.

Please, see below our reply to the reviewer's comments to Line 263-273.

Lines 43-45: please give the resolution of the boundary conditions and the format they are available in.

A reference to Section 9 (Data availability) has been added, also at the beginning of Section 5 (Global topography and bathymetry) (line 222, new numeration) and Section 6 (Global vegetation) (line 293, new numeration). Additionally, in Section 9, the format of the data has been explicited (line 617, new numeration). The resolution of the boundary condition datasets can also be found in Section 9.

Lines 52-53: the vegetation reconstruction proposed here is not exactly an update of Wolfe (1985). The sentence should be rephrased.

Done.

It has been rephrased as:

'Here, also Middle Miocene data (Pound et al., 2012; Morley, 2011) have been used'.

Section 2: a discussion explaining the use of a previous Antarctic topography corresponding to the Oligocene instead of previous published Middle Miocene topographies is needed in Section 2. Some precision could be given concerning the Oligocene configuration and how it is suitable for the Middle Miocene.

The reason why Oerlemans and Langebroek's Middle Miocene configurations were discarded is that they use rather simple model configurations. We wanted to provide a characterization of the Antarctic ice-sheet in two dimensions, i.e. varying with both latitude and longitude, and this is not available from Oerlemans or Langebroek's studies.

The scope of the current study was to provide Antarctic topography data consistent with published Antarctic ice volume estimates for the MMCO and MMG, and this was successfully accomplished. The configuration of Gasson et al. (2016) could be considered in future sensitivity studies, because uncertainties in the ice-volume estimates are high, but we consider the data from Pollard's simulations used here definitely suitable for the Middle Miocene since there is little to link those data to a specific time period (except for the Laskar orbits) (see below). Additionally, the distribution of ice in our study is comparable to that in Gasson et al. (2016): for the MMG, a continental-scale ice-sheet exists in East Antarctica with ice thicknesses of ~3000-4000 m, although in West Antarctica there is less ice in Pollard's data; for the MMCO, the ice-sheets occupy similar positions, although they are less extensive in Pollard's data.

Regarding David Pollard's simulations, the physical model used is close to that described in Pollard and DeConto (2012), but with no marine ice physics, so that any floating ice is immediately removed. The bedrock-elevation boundary conditions are from the modern ALBMAPv1 dataset (Le

Brocq et al., 2010).

Climate forcing is obtained from a matrix of previous Global Climate Model (GCM) climates for various orbits, CO2 levels and ice sizes. The GCM used is GENESIS version 3 (as in Alder et al., 2011, except with a slab mixed layer ocean). Three Earth orbits are used, with eccentricity, precession and obliquity set corresponding to warm, intermediate and cold austral summers. Three CO2 levels are used, spanning the range in the long term run. Three Antarctic ice sizes (continental, ~half and no ice cover) are specified. 10-year mean equilibrated GCM climate solutions (i.e., monthly mean surface air temperatures and precipitation) are saved for all combinations of orbit, CO2 and ice size, yielding a matrix of 27 climates.

In the long-term ice-sheet run, the appropriate climate at any point in the run is obtained by linearly weighting the surrounding saved climates in the matrix, with the weights proportional to the current austral summer insolation, ice size and logarithm of CO2 level. This matrix-forcing method is discussed further in Pollard (2010). The annual surface mass balance at each point on the ice sheet is calculated from the monthly surface air temperatures and precipitation, using a simple box (zero-dimensional) seasonal surface-mass model that includes snow storage and refreezing of meltwater, and surface melting based on Positive Degree Days (Pollard and DeConto, 2012). The run is initialized with (essentially) no ice. Insolation is based on Laskar et al. (2004). The run is 12 Myr long, nominally from "37 Ma to 25 Ma", although there is little to actually link it to specific paleodates except the Laskar orbits. From 37 to 33 Ma, CO2 decreases linearly from 6xPAL to 2xPAL. From 33 to 25 Ma, CO2 increases linearly from 2xPAL to 10xPAL (where PAL= 280 ppmv). The configuration used to represent MMCO conditions corresponds to 34.8 Ma (CO2 = 3.8xPAL). The one representing MMG conditions, to 33 Ma (CO2 = 2xPAL).

Lines 82-99 (old numeration; now lines 92-122) have been rewritten.

Section 4: the presentation of the atmospheric pCO2 estimates is rather confused. A distinction between marine and terrestrial proxy-based reconstructions of atmospheric pCO2 has to be done and discussed. Giving only the pCO2 estimates before and after the MMCT transition (corresponding to the two periods studied, MMCO and MMG) rather than the decrease throughout the transition (lines 155-161) would help to clarify the text. I also suggest adding a graph showing the pCO2 estimates in function of time in Ma. This will help to visualize the uncertainties on pCO2 estimates and the suitability of the two concentrations proposed here for MMCO and MMG.

Line 143 (old numeration; now lines 182-187) has been rewritten to indicate which studies provide marine and which terrestrial proxy-based reconstructions, and to discuss the differences in the

estimates.

We have removed lines 155-161 (old numeration) and rephrased lines 162-168 (old numeration; now lines 205-214) as follows:

'We chose atmospheric $CO_2$ concentrations of 400 ppmv and 200 ppmv to represent the MMCO and the MMG respectively (Table 1). Although somewhat arbitrary, these values are within the range of published estimates. The 400 ppmv MMCO is in favourable agreement with Foster et al. (2012) (~392 ppmv at ~15.8 Ma) and Tripati et al. (2009) (~430 ppmv at ~15.1 Ma), although higher than Pearson et al. (2000) (~300 ppmv at ~16.2 Ma) and Pagani et al. (2005) (~300 ppmv at ~15 Ma), and lower than Kürschner et al. (2008) (> ~400-500 ppmv at ~15.5 Ma) and Retallack (2009) (~852 ppmv at ~15.6 Ma) maxima. The 400 ppmv estimate is also in good agreement with the most recent alkenone- and boron isotope-based pCO2 reconstructions for the MMCO by Zhang et al. (2013) and Greenop et al. (2014). The 200 ppmv MMG estimate is in good agreement with Foster et al. (2012) (~200 ppmv at ~12 Ma) and Pagani et al. (2005) (~200 ppmv at ~13 Ma), although higher than Pearson et al. (2000) (~140 ppmv at ~14.7 Ma) and Retallack (2009) (~116 ppmv at ~14.6 Ma), and lower than Tripati et al. (2009) (~340 ppmv at ~12 Ma) and Kürschner et al. (2008) (~280 ppmv at ~14 Ma) minima.'

The reviewer's suggestion of adding a graph showing the pCO2 estimates is interesting, although we do not see the addition of a figure as a requirement for the comprehension of the CO2 section.

Line 163-164: 400 ppmv is not a maximum value of pCO2 for the MMCO if you take into account the reconstructions based on stomatal indices (Kürschner et al., 2008), pedogenic carbonates (Retallack, 2009) and recent estimates based on boron isotopes and alkenones (Foster et al., 2012).

This comment has been taken into account when rephrasing lines 162-168 (old numeration; now lines 205-214) (please, see above).

Subsection 5.3: the description of the gateway reconstruction is too detailed. I suggest putting lines 204 to 214 to the Appendix.

The description at lines 204-214 has been shortened. Nevertheless, it is important that we describe what exact modifications were applied to the dataset of Herold et al. (2008), and the South East Asian gateway is one of the modified areas and an important focus of our study. Hence, we would prefer keeping the whole description as a part of the main text.

Section 6:

Line 249: Herold et al. (2010) prescribed a vegetation distribution derived from Wolfe (1985) using a biome classification for CCSM3 adapted from Bonan et al. (2002). Did you use the same classification here? Could you please discuss the eventual differences between the classifications as they are used with the same land-surface model? I think it could be interesting to add a comparison of your MMCO vegetation reconstruction to the reconstruction proposed in Herold et al. (2010) and to highlight the differences induced by the use of the Pound et al. (2012) dataset.

Unlike Herold et al. (2010), we used the classification described in Bonan et al. (2002) (shown in Table 2 in that study) without modifying it.

Herold et al. (2010) state:
"We classify our vegetation types to a set of biomes modified from Bonan et al. (2002). These modifications include replacing C4 grass with C3 grass, since the former did not become widespread until the late Miocene, and creating a temperate broadleaf evergreen biome to more accurately represent Wolfe's (1985) middle latitude vegetation (c.f. Wolfe, 1985; Bonan et al., 2002)."

Three out of the 28 LSM biomes contain the pft "c4 grass". These biomes are "savanna" (with a 70% of "c4 grass" cover), "warm grassland" (60%), and "cool grassland" (20%) (see Table 2 in Bonan et al., 2002), which do not appear in our reconstruction (Figure 4). Hence, that modification was not required in our representation.

The regions that Herold et. al (2010) painted with the customized LSM biome "temperate broadleaf evergreen forest" (Figure 5 in that study) roughly concide with areas assigned either a) "microphyllous broadleaved evergreen forest", b) "notophyllous broadleaved evergreen forest", c) "mixed broadleaved evergreen and coniferous forest", d) "mixed broadleaved evergreen and deciduous forest", e) "mixed mesophytic forest", or f) "notophyllous woodland/xerophyllous scrub" in the reconstruction from Wolfe (1985).
The "temperate broadleaf evergreen forest" regions in the reconstruction from Herold et. al (2010) appear mostly represented by the BIOME4 biome "warm-temperate evergreen broadleaf and mixed forest" in the reconstruction from Pound et al. (2012). That BIOME4 biome represents either a) "temperate broadleaved evergreen trees" alone, or b) "cool conifer trees" mixed with "temperate

broadleaved evergreen trees", or c) "temperate deciduous trees" mixed with either "temperate broadleaved evergreen trees" or "cool conifer trees".

We converted the "warm-temperate evergreen broadleaf and mixed forest" BIOME4 biome into the LSM scheme as "warm mixed forest". The "warm mixed forest" LSM biome contains a mixture of "needleleaf evergreen temperate trees" and "broadleaf deciduous temperate trees".

We agree that the conversion is suboptimal (although the best available), because the pft "broadleaf evergreen temperate tree" is not present in the "warm mixed forest" LSM biome. However, the "warm mixed forest" LSM biome still constitutes a fair representation of the "warm-temperate evergreen broadleaf and mixed forest" BIOME4 biome and the above mentioned vegetation types from Wolfe (1985).

We agree with the reviewer that it would be interesting to compare the reconstruction in Herold et al. (2010) with Figure 4 in our study. Nevertheless, here our scope was to provide a valid reconstruction, by arguing our choice of biomes, and this was accomplished. Therefore, we would like to leave that comparison to the reader.

A note discussing the questions raised by the reviewer has been added at lines 309-320 (new numeration).

Line 263-273: I do not agree with the argument proposed here by the authors. The cooling and drying at mid-latitudes has a non-negligible impact on the vegetation distribution (as also stated by the authors in the Introduction, lines 36-37). This effect could be seen on a 2° x 2° resolution map, or even at the T42-resolution used in the CCSM3 simulations with a more detailed biome classification. This vegetation changes can in turn affect the climate-vegetation interactions (even only via the surface albedo changes) and significantly impact on the global climate. I suggest at least revising the vegetation distribution for the MMG and to detail the biome classification used here in order to better represent the changes between MMCO and MMG vegetation distributions (see General Comment 1).

We agree with the reviewer that the appearance of cooler and drier biomes at mid-latitudes during the Serravallian could have an effect on global climate. Nevertheless, Pound et al. (2012) state that despite these changes the vegetation patterns of the Langhian and Serravallian were "similar" (please, see Figure 5 in Pound et al., 2012), which contrasts with the "markedly different biome pattern of the Tortonian from that of the Serravallian" (please, compare Figure 5 and Figure 6 in Pound et al., 2012).

A clear change in mid-latitude biomes from the Langhian to the Serravallian is visible only in two

areas (Figure 5 in Pound et al., 2012): western North America and Europe. During the Serravallian, in the western North American mid-latitudes the "warm-temperate evergreen broadleaf and mixed forest" ("warm mixed forest" in LSM scheme) was still present but other drier/cooler biomes such as "temperate deciduous broadleaf forest" ("warm broadleaf deciduous forest" in LSM scheme) or "cool-temperate mixed forest" ("cool mixed forest" in LSM scheme) had appeared (Pound et al., 2012). In Europe, evidence of cooling/drying during the Serravallian comes from one site in central Spain representing "temperate sclerophyll woodland" ("evergreen shrub land" in LSM scheme), two sites in southern France representing "temperate deciduous broadleaf savanna" ("deciduous shrub land" in LSM scheme), "temperate deciduous broadleaf forest" ("warm broadleaf deciduous forest" in LSM scheme) in southern Germany, and three sites east of 28°E indicating "temperate deciduous broadleaf savanna". Nevertheless, the "warm-temperate evergreen broadleaf and mixed forest" ("warm mixed forest" in LSM scheme) continued to be the main biome in Europe during the Serravallian (Pound et al., 2012).

Thus, for studies with a specific focus on vegetation triggered climatic changes across the MMCT, the user could modify our MMG vegetation dataset (LSM scheme) as follows (based on Pound et al., 2012):

a) In the mid-latitudes of western North America, the "warm mixed forest" between 40-50°N could be partly replaced with "warm broadleaf deciduous forest". Also a "cool mixed forest" could be added in the same region at 42°N.

 b) In Europe, some "deciduous shrub land" could be added to the "warm mixed forest" in southern France between 42.5-44°N and 6-9°E, and also between 38-47°N and 29-36°E.

This point is discussed at lines 339-346 (new numeration).

Lines 273-274: how much does the Miocene vegetation distribution differ from the pre-industrial vegetation distribution, as used in CCSM3. It can be useful to briefly list the differences here to better highlight the potential impact of vegetation on the Middle Miocene climate if using the boundary condition set proposed here. I also suggest adding a figure showing the PI vegetation distribution with the same biome classification (maybe in Figure 4).

We have added the following text at lines 350-355 (new numeration):
'Compared to PI, the vegetation of the Middle Miocene represents a warmer and wetter climate. In the northern hemisphere high latitudes forests are warmer, with no forest tundra or tundra present. The mid-latitudes present warmer and wetter biomes, with e.g. less shrub land type biomes. The tropics are wetter, with less savanna and less grasses. There is no evidence for neither a desert in

northern Africa (Sahara) nor in central Asia. In the southern hemisphere high latitudes tundra is present at the MMCO and disappears after the Antarctic ice-sheet expansion at the MMG (Pound et al., 2012; Bonan et al., 2002).'

The reader could check Figure 6 in Bonan et al. (2002) for a comparison with modern vegetation in the LSM scheme.

Subsections 6.1 to 6.9: I suggest making these subsections more concise. I would prefer to have only one paragraph focusing on the major vegetation patterns that are taken here into account for the MMCO and MMG. The detailed description of regional vegetation patterns is useless because most of them are neglected for simplification. The authors can directly refer to Pound et al. (2012) for more detailed information.

Although a detailed discussion on the vegetation of each region is not indispensable for the comprehension of this manuscript, we think that it is important to show how exactly we decided what vegetation to assign to each region.

Subsections 6.1-6.9 have thus been moved to the appendix (Section 10; line 622 new numeration), in case the reader was interested in those details.

Section 7:

Lines 467-475: the presentation and discussion of simulation results need to be reworked and extended. What are the global mean surface air temperature and precipitation differences between the two Miocene runs and the PI run? How do you explain that the MMG run is warmer than the PI run? Is it linked to the absence of ice in the Northern Hemisphere? What is the contribution of the boundary condition changes to the climate differences that the model simulates? A brief comparison with previous modeling studies is highly welcome here. A comparison with some proxy-data (e. g. for SSTs) can also be added.

The global mean surface air temperatures (at 2 m height) are 16.38°C, 13.88°C, and 12.16°C for the MMCO, MMG, and PI experiments, respectively. The global mean precipitation rates are 3.00, 2.86, and 2.72 mm/day for the MMCO, MMG, and PI experiments, respectively.

Potential causes for a MMG climate warmer than PI could be the lower extent of ice-sheets (the Antarctic ice-sheet is smaller and the northern Hemisphere free of ice-sheets in the MMG run), or

the different vegetation cover (Knorr et al., 2011). However, unambiguously disentangling the effects of each of the different boundary conditions would require performing a series of sensitivity experiments, which was beyond the scope of the current study. Here our aim was testing the idoneity of the current boundary conditions as input data in GCMs for MMCO and MMG experiments.

Our global mean surface air temperature and precipitation values support the idea of a Middle Miocene climate warmer and wetter than PI, and a cooling and drying trend across the MMCT, as suggested e.g. in Pound et al. (2012).

Mg/Ca data from ODP Hole 1171C on the South Tasman Rise indicate cooling of SST's of ~2°C across the MMCT (Shevenell et. al, 2004). This value is within our range of cooling estimates for the Southern Ocean.

Knorr and Lohmann (2014) MMCT model results show a decrease of 3.1°C in global mean surface air temperature across the MMCT, a value slightly higher than our 2.5°C estimate.

The questions raised by the reviewer have been addressed at lines 570-598 (new numeration).

Concluding remarks: this section needs to be reworked in function of the amendments of the previous sections.

**Figures and tables:**

Figure 5: I would suggest adding maps of mean surface air temperature differences (MMCO and MMG-PI). It could also be interesting to show the temperature differences between MMCO and MMG.

A map of surface air temperature differences (MMCO-PI and MMG-PI) has been added.

[Figure]

**Figure S3: Surface air temperature (at 2 m height) (°C) differences between MMCO and MMG experiments, and PI, respectively.**

Table 2: is the correspondence between cool-temperate mixed forest (BIOME 4) and cool mixed forest (LSM) really suitable, since you mention in the footnotes that the cool mixed forest represents only boreal trees? Isn't it another possibility of correspondence?

The "cool-temperate mixed forest" biome represents either a) a forest dominated by "boreal evergreen trees" but with also "temperate deciduous trees" present and a coldest month temperature > -19°C or b) a forest dominated by "temperate deciduous trees" but with also "boreal evergreen trees" present and a coldest month temperature > -15°C.
The "cool mixed forest" biome represents a mixture of "needleleaf evergreen boreal trees" and "broadleaf deciduous boreal trees".
We agree with the reviewer that the correspondence is not optimal, because the deciduous trees in the "cool-temperate mixed forest" are temperate, meanwhile the ones in the "cool mixed forest" are boreal . Nevertheless, there is not a better possibility of correspondence since all the cool forests in the LSM scheme contain only boreal trees (and all the warm forests contain only temperate trees).

Table 3: could you please give explicitly the values of the model parameters instead of citing a reference paper? Same for the PI orbital parameters.

Done. Please, see the updated table below.

| Experiment | PI | MMCO | MMG |
|---|---|---|---|
| $CO_2$ | 280 ppmv | 400 ppmv | 200 ppmv |
| $CH_4$ | 760 ppbv | | |
| $N_2O$ | 270 ppbv | | |
| CFC's | 0 | | |
| $O_3$ | 1870 A.D. | | |
| Sulfate aerosols | 1870 A.D. | | |
| Dust and sea salt | PD | same as PI | |
| Carbonaceous aerosols | 30% of PD | | |
| Solar constant | 1365 $Wm^{-2}$ | | |
| Eccentricity | 0.016724 | | |
| Obliquity | 23.446 ° | | |
| Precession | 102.04 ° | | |

**Table 3: Summary of atmospheric composition, solar constant, and orbital configuration for the CCSM3 test experiments. PI values are according to Otto-Bliesner et al. (2006). The orbital configuration represents 1950 A.D. values. PD = present day.**

Further details on the ozone and sulfate aerosols distribution can be found in Otto-Bliesner et al. (2006).

**Technical comments:**

-Line 50: replace "passages" by "seaways"

Replaced.

-Line 51: add the precision "most previous Middle Miocene studies with prescribed vegetation"

Added.

-Lines 56-57: could you please rephrase this sentence? There are other ways to produce boundary condition assemblages.

Done.
Rephrased as:
'Despite the relatively low availability of Middle Miocene data'.

-Line 93: replace "6 estimate" by "volume estimate"

Replaced (now line 95).

-Line 133: write "previous Section"

The journal guidelines under https://www.geoscientific-model-development.net/for_authors/manuscript_preparation.html suggest writing Sect. instead of Section, unless that word appears at the beginning of a sentence.
The instructions state literally:
'The abbreviation "Sect." should be used when it appears in running text and should be followed by a number unless it comes at the beginning of a sentence.'

-Line 165: delete the space between "p" and "CO2"

Done.

-Line 178: "ice-free conditions"

Done. "ice free" has been replaced with "ice-free", at line 178 and, for consistency, in all other occcurrences in the text.

-Line 191: replace "passages" by "seaways"

Done.

-Line 194: write "Section 2":

Please, see above our answer to the comment to Line 133.

-Line 230: could you please use "seaway" instead of passage or Central American seaway.

Done.
"Panama passage" has been replaced with "Panama seaway". Also at Line 231 (old numeration).

-Line 312: "Northeast Australia"

The following is stated in the journal guidelines under
https://www.geoscientific-model-development.net/for_authors/manuscript_preparation.html:

'Cardinal directions should only be capitalized when part of a proper noun (e.g. South Dakota, Northern Ireland, North America, but eastern France).'

-Lines 318, 320: "East Australia"

Please, see above.

-Line 322 and after: I always put a caption letter to subregions or continents "West Australia", "Southern Africa", etc.

Please, see above.

-Line 448: please explain configuration T42x1 or detail.

T42 is the atmosphere horizontal grid, a Gaussian grid with 64 points in latitude and 128 points in longitude (~2.8° resolution). The notation T42 refers to the spectral truncation level. x1 is the ocean horizontal grid, a dipole grid with 384 points in latitude and 320 points in longitude. The zonal resolution of the ocean horizontal grid is ~1°, the mean meridional resolution is ~0.5°, refined around the equator (~0.3°). The notation x1 refers to the nominal zonal resolution. T42x1 is the model configuration employing the T42 and x1 grids.

Lines 448-453 (old numeration) have been modified as follows:
'The atmosphere horizontal grid employed in the PI run, T42, is a Gaussian grid with 64 points in latitude and 128 points in longitude (~2.8° resolution). The notation T42 refers to the spectral truncation level. The land and atmosphere models share the same horizontal grid. The ocean horizontal grid, x1, is a dipole grid with 384 points in latitude and 320 points in longitude. The zonal resolution of the ocean horizontal grid is ~1°, the mean meridional resolution is ~0.5°, refined around the equator (~0.3°). The notation x1 refers to the nominal zonal resolution. The ocean and sea–ice components share the same horizontal grid. The atmosphere and ocean vertical grids have 26 and 40 vertical levels, respectively. This model grid configuration is known as T42x1.'

-Line 464: "archived as b30.043" does this information really need to be mentioned?

It is not indispensable. We removed it.

**Additional modifications:**

-Line 51 (now line 59): "were mainly based" was gramatically incorrect. It was replaced with "was mainly based".

-Line 237 (now line 295): 15.67 has been replaced with 15.97. The Langhian expands the interval 15.97– 13.65 Ma

-Lines 460-463 (now lines 559-560): The orbital configuration used in the Miocene experiments is identical to the one used in the PI experiment. There was a mistake in our statement there, sorry about that. Those lines were rephased as follows to correct the mistake:

'Well-mixed greenhouse gases, ozone, aerosols, solar constant and orbital configuration were kept the same as in PI, except for $CO_2$ (Table 3).'

-Line 493 (now line 618): we replaced "2°x2° lat/lon grid" with "0.5°x0.5° lat/lon grid". Although Herold et al. (2008) topography/bathymetry dataset was provided to us in a 2°x2° resolution, we regridded it to a finer resolution (0.5°x0.5°) for our purposes.

-Lines 492-495 (now lines 616-621): a reference to the CCSM3 model output files from the MMCO, MMG, and PI experiments included in the supplement has been added.

We hope we have addressed all your comments.

Yours sincerely,

Amanda Frigola and co-authors.

**References:**

Alder, J. R., Hostetler, S. W., Pollard, D., and Schmittner, A.: Evaluation of a present-day climate simulation with a new coupled atmosphere-ocean model GENMOM, GMD, 4(1), 69–83, doi:10.5194/gmd-4-69-2011, 2011.

[revised manuscript text omitted]